# Online Learning of Delayed Choices

**Recep Yusuf Bekci**
University of Waterloo
Waterloo, Canada
`recep.bekci@uwaterloo.ca`

## Abstract

Choice models are essential for understanding decision-making processes in domains like online advertising, product recommendations, and assortment optimization. The Multinomial Logit (MNL) model is particularly versatile in selecting products or advertisements for display. However, challenges arise with unknown MNL parameters and delayed feedback, requiring sellers to learn customers' choice behavior and make dynamic decisions with biased knowledge due to delays. We address these challenges by developing an algorithm that handles delayed feedback, balancing exploration and exploitation using confidence bounds and optimism. We first consider a censored setting where a threshold for considering feedback is imposed by business requirements. Our algorithm demonstrates a $\tilde{O}(\sqrt{NT})$ regret, with a matching lower bound up to a logarithmic term. Furthermore, we extend our analysis to environments with non-thresholded delays, achieving a $\tilde{O}(\sqrt{NT})$ regret. To validate our approach, we conduct experiments that confirm the effectiveness of our algorithm.

## 1 Introduction

The ability to model and understand consumer choices between discrete alternatives is critical for various business applications, such as online advertising, product recommendations, and assortment optimization. Businesses need to present the most appealing set of options to consumers to maximize engagement and revenue. However, the task of optimizing what content or products are shown to a customer during their browsing session is complex due to the interplay between alternatives that customers face. Each alternative can act as a substitute or competitor to others, impacting the customer's final decision. Traditional multi-armed bandit (MAB) models, which are widely used for decision-making problems, fall short in scenarios where a subset of alternatives must be presented, and the customer's choice among these influences future decisions.

Multinomial choice (MNL) models have emerged as powerful tools for capturing and predicting consumer behavior among a finite set of alternatives. These models estimate the utilities of different options and the probabilities of their selection. However, when the MNL parameters are unknown and no historical data is available—as is often the case with newly introduced products or advertisements—the learning process becomes even more challenging. This complexity is further exacerbated when the feedback on decisions is delayed, requiring the learner to dynamically adjust decisions based on limited and potentially biased information.

One of the fundamental challenges in this setting is the delay in receiving feedback from customers. Unlike immediate responses in classical MAB problems, customers in e-commerce environments often take hours or even days to make decisions, as highlighted by Vernade et al. [2020] and Chapelle [2014]. This delay in feedback complicates the learning process, as it must adapt to new information that arrives sporadically and potentially long after the initial interaction.

38th Conference on Neural Information Processing Systems (NeurIPS 2024).

In this paper, we address the dual challenges of unknown MNL parameters and delayed feedback by developing algorithms that balances exploration and exploitation through the use of confidence bounds and the principle of optimism in the face of uncertainty. We focus on two settings in receiving delay: the thresholded and the non-thresholded settings.

In thresholded settings, feedback is only considered if it is received within a predetermined time frame set by business requirements. This constraint ensures operational stability and efficiency by ignoring excessively delayed responses that may no longer be relevant. For these settings, we introduce the Delayed Multinomial Logit Bandit (DEMBA) algorithm, specifically designed to handle potentially censored delayed feedback effectively.

In contrast, non-thresholded settings allow the learner to consider all feedback regardless of delay, potentially leading to better accuracy in decision making in long-term but at the cost of increased bias in the learning process. For such environments, we propose the Patient Delayed Multinomial Logit Bandit (PA-DEMBA) algorithm, which adapts its learning strategy to accommodate all feedback, irrespective of the delay.

**Contributions.** Our main contributions are two novel bandit algorithms: we develop DEMBA, for thresholded feedback settings, and PA-DEMBA, an algorithm designed for non-thresholded feedback environments. Both algorithms effectively learn from delayed and censored choices using confidence intervals. We provide a comprehensive regret analysis for both DEMBA and PA-DEMBA, demonstrating an $\tilde{O}(\sqrt{NT})$ regret bound and a matching lower bound up to a logarithmic term. Additionally, through detailed computational experiments, we validate the performance and robustness of our algorithms in various scenarios.

**Organization.** The remainder of this paper is organized as follows. In Section 2, we review the related work on choice modeling and delayed feedback in online learning. Section 3 details the problem formulation and the specific challenges addressed by our approach. Our main algorithm DEMBA is presented in Section 4. We analyze the regret bounds of this algorithm in Section 5, providing theoretical guarantees for its performance. PA-DEMBA algorithm for non-thresholded delays is presented in Section 6. In Section 7, we conduct experiments to demonstrate the effectiveness of our proposed algorithms. Finally, Section 8 concludes the paper and outlines potential directions for future research.

## 2 Related Work

Delayed feedback is a crucial aspect of online learning environments, especially in domains like online advertising and e-commerce, where decision making involves a consideration period, or in healthcare, where the effects of actions take time to manifest. Consequently, research interest in bandits with delayed feedback has surged in recent years.

In their seminal work, Joulani et al. [2013] studied online learning scenarios with stochastic delayed feedback. Their work laid the foundation for understanding how delays impact learning performance. Similarly, Chapelle [2014] examined delayed conversions in display advertising, highlighting the practical challenges faced in real-world applications. Further, Vernade et al. [2017] focused on delays specifically in the context of delayed conversions, providing insights into handling delays with known distributions.

Expanding on this, Pike-Burke et al. [2018] explored bandits with delayed, aggregated, and anonymous feedback, which adds another layer of complexity by considering multiple types of delays. Zhou et al. [2019] extended this exploration to generalized linear contextual bandits with delayed feedback. Vernade et al. [2020] investigated linear bandits with stochastic thresholded delays. Additionally, Gael et al. [2020] tackled multi-armed bandits with arm-dependent stochastic delays, focusing on the challenges of non-uniform delays across different choices. Moreover, Cesa-Bianchi et al. [2022] considered composite and anonymous delayed feedback within non-stochastic bandits, further enriching the literature on delayed feedback mechanisms. Tang et al. [2024] studied delayed multi-armed bandits (MAB) with reward-dependent delays in clinical trials, while Lancewicki et al. [2021] explored both reward-dependent and reward-independent delay settings. Flaspohler et al. [2021] investigated delayed learning in weather forecasting, and Grover et al. [2018] addressed the best arm identification problem with delayed feedback. Thune et al. [2019] examined non-stochastic MABs with unrestricted delays, and Cesa-Bianchi et al. [2016] considered cooperation between

different agents in delayed settings. Tang et al. [2021] explored scenarios where past actions impact future arm rewards, and Yang et al. [2024] addressed general sequential decision-making problems with delayed feedback. Despite this rich body of work, the solutions developed for MABs do not directly apply to our setting, where assortment feedback in discrete choice models presents additional complexities. In particular, delayed feedback affects both item value estimation and assortment composition, making our problem significantly more challenging than those where only arm rewards are updated.

When focusing specifically on generalized linear bandits with delays, we note key contributions such as Howson et al. [2023] and Blanchet et al. [2024] who explored generalized linear bandits with delayed feedback, demonstrating the efficacy of these models in more complex, non-linear settings. Multinomial bandits, which address decision making where multiple items are offered simultaneously, present a unique challenge due to the interactions between items. This complexity distinguishes our problem from other online learning models. Specifically, for generalized linear bandits, we note that when the assortment has more than one item, our problem cannot be addressed by solutions designed for generalized linear models due to the complexity of interactions among multiple choices which makes the action space more complicated.

In terms of online learning with choice models, significant progress has been made in understanding and optimizing MNL parameters. Agrawal et al. [2017] developed a Thompson sampling algorithm for learning MNL parameters for assortment optimization, while Agrawal et al. [2019] proposed a UCB algorithm for the same purpose. Dong et al. [2020] adapted this problem by introducing switch costs, addressing practical constraints in dynamic environments. Further, Agarwal et al. [2020] studied the problem for best arm identification, extending the framework to multiple pulls, which is a generalization of the dueling bandits problem. Other notable works include Wang et al. [2018] and Chen et al. [2021], who worked on dynamic assortment allocation under uncapacitated MNL models, and Perivier and Goyal [2022], who investigated joint pricing and assortment optimization with MNL demand processes.

To the best of our knowledge, our work is the first to address online learning of delayed choices in this context. Specifically, we propose the Delayed Multinomial Logit Bandit (DEMBA) algorithm for thresholded feedback settings and the Patient Delayed Multinomial Logit Bandit (PA-DEMBA) algorithm for non-thresholded settings. These algorithms effectively learn from delayed and censored choices using confidence intervals, providing robust solutions for dynamic and uncertain environments where feedback is not immediately available. Our approach not only advances the theoretical understanding of delayed feedback in online learning but also offers practical insights for applications in e-commerce and online advertising.

# 3 Problem Setup

We consider a capacitated selection problem faced by a *seller* over $T$ rounds. The items to be selected, referred to as *products*, can be new retail products or services such as advertisements. There are $N$ products available to potentially be shown to the *customer*. The set of products selected in a given round is called an *assortment*, denoted as $S_t$ for round $t$, where $S_t \subset \{1, \ldots, N\}$ and $|S_t| \leq K$. Here, $K$ represents the capacity, indicating the maximum number of items the seller can show at any given time.

Upon encountering the assortment, the customer makes a choice $a_t$ among the options: (i) rejecting the browsing options provided ($a_t = 0$), (ii) browsing and purchasing/selecting an option ($a_t = i, i \in S_t$), and (iii) browsing but not purchasing/selecting an option ($a_t = 0d$). If option $i$ is chosen, the seller earns a reward $r_i$, with $r_i \in [0, 1]$ for $i \in [N]$ and $r_0 = r_{0d} = 0$.

Customer choice probabilities are determined by the Multinomial Logit (MNL) model as follows:

$$\mathbb{P}(a_t = i | S_t = S) = \begin{cases} \frac{v_i}{v_0 + v_{0d} + \sum_{j \in S} v_j} & if \ i \in S \cup \{0, 0d\} \\ 0 & otherwise, \end{cases}$$

where $v_i$ denotes the (unknown) attraction parameter of option $i$. Without loss of generality we assume that attraction parameters are normalized such that $v_0 = 1$. In parallel with the real applications, we also assume that not browsing is the most common choice, i.e. $v_i \leq v_0$.

It is important to note that not purchasing/selecting is different from tracking conversions. Specifically, if the customer browses the given assortment, we can track if the customer decides not to purchase, similar to choosing an option to purchase (e.g., by closing the pop-up or following specific behavioral click-through patterns).

If the customer's choice is $a_t = 0$, the seller receives this choice immediately. Otherwise, the customer's choice is revealed to the seller after a delay $d_t \in \mathbb{N}$. $d_t$ is sampled from a unknown distribution $f_d$ and independent from $a_t$. Moreover, delays longer than a certain threshold $\mu$ is censored or ignored by the seller in the learning process. This threshold is set based on the seller's operational requirements. We later extend our solution to the *patient learner* setting without a threshold, i.e. $\mu \to \infty$.

We define $a_{i,t}$ as the demand for option $i$ at time $t$. We have

$$a_{i,t} = \begin{cases} 1, & if \ a_t = i, \\ 0, & otherwise. \end{cases}$$

We also define $c_{i,s,t} \in \{0, 1\}$ as the censoring variable of product $i$ at period $t$ that is sold at period $s$. The censoring variable is determined as:

$$c_{i,s,t} = \mathbb{I}(d_s \leq t - s \text{ and } d_s \leq \mu).$$

We define the feedback observed by the seller as $o_{i,s,t} \in \{0, 1\}$ and $o_{i,s,t} = c_{i,s,t} a_{i,s}$. The expected fraction of observed feedback is denoted as $\psi_\mu := \sum_{s=0}^{\mu} f_d(s)$.

The sequence of events at round $t$ can be summarized as follows:

1. The seller selects an assortment $S_t$.
2. The customer interacts with the medium(view the page or encounter with the pop-up) and makes a decision $a_t$.
3. The environment returns a reward $r_i$, $i \in [N]$, and samples a delay $d_t$.
4. Rewards of certain previous actions and/or if the customer rejected to browse revealed to the seller as $o_{i,s,t}$.

The expected reward of the seller given assortment $S$ and attraction parameter set $v$ is given by

$$R(S; v) = \sum_{i \in S} r_i \cdot \frac{v_i}{1 + v_{0d} + \sum_{j \in S} v_j}.$$

The goal of the seller is to sequentially learn customer preferences and find a policy to minimize cumulative expected regret, defined as:

$$\text{Reg}(T, \pi) = \sum_{t=1}^{T} R(S^*; \psi_\mu v) - R(S_t^\pi; \psi_\mu v),$$

where $S^* = \arg\max_{S \subset \{1,\ldots,N\}, |S| \leq K} R(S; \psi_\mu v)$ maximizes the expected reward of the clairvoyant and $v = \{v_{0d}, v_1, \ldots, v_N\}$ is the ground truth attraction parameter set.

## 4 Delayed MNL Bandit (DEMBA) Algorithm

In this section, we introduce the Delayed MNL Bandit (DEMBA) algorithm. DEMBA leverages an epoch-based learning method, where epochs are explicitly defined by immediate no-purchase decisions. Specifically, when a no-purchase decision is made by the customer, the current epoch is closed, and a new one begins. Throughout each epoch, customer selections are observed, and parameter updates occur upon encountering a no-purchase outcome.

Our approach adopts the principle of optimism in the face of uncertainty [Auer et al., 2002] for parameter estimation, generating optimistic estimations using upper confidence bounds for product attraction parameters and a lower confidence bound for the no-purchase option. This results in an optimistic revenue function, guiding decision-making under uncertainty by balancing exploration and exploitation.

We build our optimistic estimates on biased observations. The total observed preference given to product $i$ until epoch $\tau$ is denoted as $\tilde{v}_{i,\tau}$ and can be calculated as

$$\tilde{v}_{i,\tau} = \sum_{s=1}^{t_\tau^{end}} o_{i,s,t_\tau^{end}},$$

where $\tau$ is the current epoch and $t_\tau^{end}$ is the last period of epoch $\tau$.

We count how many times a particular product $i$ is offered in the assortment until epoch $\tau$ using the set $E_\tau(i)$ which is the set of epochs during which $i$ is offered, i.e. $E_\tau(i) = \{e \leq \tau : i \in S_e\}$. Then, we estimate attraction parameters by

$$\hat{v}_{i,\tau} = \frac{\tilde{v}_{i,\tau}}{|E_\tau(i)|}.$$

$\hat{v}_{i,\tau}$ is a biased estimator due to delay and thresholding, i.e. $\mathbb{E}[\hat{v}_{i,\tau}] \neq v_i$. We consider this bias in our estimation and build our concentration around $\psi_\mu v_i$. We state our concentration result in the following lemma.

**Lemma 4.1** *With probability at least $1 - O(N^{-2}T^{-1})$, for every epoch $\tau \in \{1, \dots\}$ and option $i \in \{0d, 1, \dots, N\}$, we have*

$$|\hat{v}_{i,\tau} - \psi_\mu v_i| \leq \Delta_{i,\tau},$$

*where $\Delta_{i,\tau} = \sqrt{\frac{(48\hat{v}_{i,\tau}+1)\log(NT)}{|E_\tau(i)|}} + \frac{48\log(NT)+\mu}{|E_\tau(i)|}$.*

*Sketch of the proof.* Note that $\hat{v}_{i,\tau}$ is a biased estimator and defined as

$$\hat{v}_{i,\tau} = \frac{\tilde{v}_{i,\tau}}{|E_\tau(i)|} = \frac{\sum_{s=1}^{t_\tau^{end}} o_{i,s,t_\tau^{end}}}{|E_\tau(i)|},$$

where $|E_\tau(i)|$ is the total number of epochs during which product $i$ is shown to the customer and $\sum_{s=1}^{t_\tau^{end}} o_{i,s,t_\tau^{end}}$ is the total observed sales of product $i \neq 0d$ or is the total observed delayed no selections.

We analyze the concentration around $\psi_\mu v_i$:

$$
\begin{aligned}
|\hat{v}_{i,\tau} - \psi_\mu v_i| &= \left| \frac{\sum_{s=1}^{t_\tau^{end}} o_{i,s,t_\tau^{end}}}{|E_\tau(i)|} - \psi_\mu v_i \right| \\
&= \left| \frac{\sum_{s=1}^{t_\tau^{end}-\mu} a_{i,s}\mathbb{I}(d_s \leq \mu) + \sum_{s=t_\tau^{end}-\mu+1}^{t_\tau^{end}} a_{i,s}\mathbb{I}(d_s \leq t_\tau^{end} - s)}{|E_\tau(i)|} - \psi_\mu v_i \right| \\
&= \left| \frac{\sum_{s=1}^{t_\tau^{end}} a_{i,s}\mathbb{I}(d_s \leq \mu) + \sum_{s=t_\tau^{end}-\mu+1}^{t_\tau^{end}} a_{i,s}\left(\mathbb{I}(d_s \leq t_\tau^{end} - s) - \mathbb{I}(d_s \leq \mu)\right)}{|E_\tau(i)|} - \psi_\mu v_i \right| \\
&\leq \left| \frac{\sum_{s=1}^{t_\tau^{end}} a_{i,s}\mathbb{I}(d_s \leq \mu)}{|E_\tau(i)|} - \psi_\mu v_i \right| + \left| \frac{\sum_{s=t_\tau^{end}-\mu+1}^{t_\tau^{end}} a_{i,s}\left(1 - \mathbb{I}(d_s \leq \mu)\right)}{|E_\tau(i)|} \right|, \quad (1)
\end{aligned}
$$

where the second equality follows from decomposing the observations into those before and after the threshold, the third equality rearranges the terms, and the last inequality uses the triangle inequality.

For the first term in the decomposition (1), we have

$$\left| \frac{\sum_{s=1}^{t_\tau^{end}} a_{i,s} \mathbb{I}(d_s \leq \mu)}{|E_\tau(i)|} - \psi_\mu v_i \right| = \left| \frac{\sum_{s=1}^{t_\tau^{end}} a_{i,s} \mathbb{I}(d_s \leq \mu)}{|E_\tau(i)|} - \frac{\psi_\mu \sum_{s=1}^{t_\tau^{end}} a_{i,s}}{|E_\tau(i)|} + \frac{\psi_\mu \sum_{s=1}^{t_\tau^{end}} a_{i,s}}{|E_\tau(i)|} - \psi_\mu v_i \right|$$

$$\leq \left| \frac{\sum_{s=1}^{t_\tau^{end}} a_{i,s} \mathbb{I}(d_s \leq \mu)}{|E_\tau(i)|} - \frac{\psi_\mu \sum_{s=1}^{t_\tau^{end}} a_{i,s}}{|E_\tau(i)|} \right| + \left| \frac{\psi_\mu \sum_{s=1}^{t_\tau^{end}} a_{i,s}}{|E_\tau(i)|} - \psi_\mu v_i \right|$$

$$\underbrace{\phantom{xxxxxxxxxxxxxxxxxxxxxxxxxx}}_{(a)} \quad \underbrace{\phantom{xxxxxxxxxxxxxxxxx}}_{(b)}$$

$$\leq \underbrace{\left| \frac{\sum_{s=1}^{t_\tau^{end}} \mathbb{I}(d_s \leq \mu)}{|E_\tau(i)|} - \psi_\mu \right|}_{} + \underbrace{\left| \frac{\sum_{s=1}^{t_\tau^{end}} a_{i,s}}{|E_\tau(i)|} - v_i \right|}_{}. \tag{2}$$

We bound the first term $(a)$ using Hoeffding's inequality:

$$(a) \leq \sqrt{\frac{\log(NT)}{|E_\tau(i)|}}. \tag{3}$$

For part $(b)$, we make use of the Chernoff bound from Theorem A.1 and handle two cases based on $\zeta = (v_i + 1)\sqrt{\frac{6\log(NT)}{v_i |E_\tau(i)|}}$. The detailed proof is provided in Appendix A. The result can be summarized as follows:

$$\mathbb{P}\left( \left| \frac{\sum_{s=1}^{t_\tau^{end}} a_{i,s}}{|E_\tau(i)|} - v_i \right| \leq \sqrt{\frac{48\hat{v}_{i,\tau} \log(NT)}{|E_\tau(i)|}} + \frac{48\log(NT)}{|E_\tau(i)|} \right) \geq 1 - \frac{4}{N^2 T}. \tag{4}$$

Combining the bounds for both terms, we establish the concentration result stated in Lemma 4.1. □

Using the concentration result for the attraction parameters, we construct upper confidence bounds for each product at each epoch:

$$\bar{v}_{i,\tau} = \hat{v}_{i,\tau} + \Delta_{i,\tau},$$

and for the delayed no-purchase option, we construct a lower confidence bound:

$$\underline{v}_{0d,\tau} = \hat{v}_{0d,\tau} - \Delta_{0d,\tau},$$

where

$$\Delta_{i,\tau} = \sqrt{\frac{(48\hat{v}_{i,\tau} + 1)\log(NT)}{|E_\tau(i)|}} + \frac{48\log(NT) + \mu}{|E_\tau(i)|}.$$

We use the optimistic parameter estimations to construct an optimistic revenue function

$$R(S; \bar{v}) = \sum_{i \in S} r_i \frac{\bar{v}_{i,\tau}}{1 + \underline{v}_{od,\tau} + \sum_{j \in S} \bar{v}_{j,\tau}}.$$

Our algorithm DEMBA suggests the assortment according to the optimistic revenue function $R(S; \bar{v})$ and updates parameters according to feedback received with delay. The pseudocode of DEMBA is given in Algorithm 1.

**Computational complexity.** The computational complexity of the DEMBA algorithm involves several key components. The most intensive step is computing the assortment $S_\tau$ by maximizing the revenue function $R(S; \bar{v})$. For this step, polynomial-time solutions with $O(N^2)$ complexity are available, as demonstrated by Rusmevichientong et al. [2010] and Davis et al. [2013]. Updating observed preferences $\tilde{v}_{i,\tau}$ involves summing over previous observations, with a complexity of $O(N(t_\tau^{start} - t_\tau^{end}))$. The process of updating sets $E_\tau(i)$ and estimations $\hat{v}_{i,\tau}$ and the confidence bounds adds $O(N)$ operations per epoch. Given that $\tau \leq t$ and $t \leq T$, the overall computational complexity across all rounds $T$ is $O(TN^2 + NT^2)$.

**Algorithm 1** Delayed MNL Bandit (DEMBA)

---

**Initialize:** $t = 0$, $\tau = 0$;
**while** $t < T$ **do**
    Compute $S_\tau = \arg\max_{S \subset \{1,\dots,N\}, |S| \leq K} R(S; \bar{v})$;
    Offer assortment $S_\tau$;
    Receive feedback;
    **if** $a_t = 0$ **then** {immediate reject}
        $\tilde{v}_{i,\tau} = \sum_{s=1}^{t_\tau^{end}} o_{i,s,t_\tau^{end}}$   $\forall i \in [N] \cup \{0d\}$;
        $E_\tau(i) = \{e \leq \tau : i \in S_e\}$   $\forall i \in [N] \cup \{0d\}$;
        $\hat{v}_{i,\tau} = \frac{\tilde{v}_{i,\tau}}{|E_\tau(i)|}$   $\forall i \in [N] \cup \{0d\}$;
        $\bar{v}_{i,\tau} = \hat{v}_{i,\tau} + \Delta_{i,\tau}$   $\forall i \in [N]$;
        $\underline{v}_{0d,\tau} = \hat{v}_{0d,\tau} - \Delta_{0d,\tau}$
        $\tau = \tau + 1$
    **end if**
    $t = t + 1$
**end while**

---

## 5 Regret Analysis

Our main result is given in the following theorem.

**Theorem 5.1** *Let* $\pi^{DEMBA}$ *be the policy produced by Algorithm 1 using* $\Delta_{i,\tau} = \sqrt{\frac{(48\hat{v}_{i,\tau}+1)\log(NT)}{|E_\tau(i)|}} + \frac{48\log(NT)+\mu}{|E_\tau(i)|}$. *Then,* $\pi^{DEMBA}$ *satisfies*

$$
\begin{aligned}
Reg(T, \pi^{DEMBA}) \leq &(1 + \mu)\log(T) + K\sqrt{73T\log(NT)} + 48K\log(T)(\log(NT) + \mu) \\
&+ 73\sqrt{NT\log(NT)} + 48\log^2(NT).
\end{aligned}
$$

*The bound can be further simplified to* $\tilde{O}\left(\sqrt{NT}\right)$ *by omitting logarithmic terms.*

*Sketch of the proof.* The proof of Theorem 5.1 consists of several steps. First, we use the definition of the optimistic revenue function $R(S; \bar{v})$ and show that it provides an upper bound on the true revenue function $R(S; \psi_\mu v)$. This is achieved by leveraging Lemma B.1, which ensures that the estimated attraction parameters are close to their true values with high probability.

We start by expressing the regret in terms of the epochs:

$$
\text{Reg}(T, \pi) = \mathbb{E}\left[\sum_{\tau=1}^{\bar{\tau}} |\mathcal{H}_\tau| \left(R(S^*; \psi_\mu v) - R(S_\tau; \psi_\mu v)\right)\right],
$$

where $\mathcal{H}_\tau$ is the duration of epoch $\tau$. Given that the epoch duration follows a geometric distribution, we simplify the expected regret using the law of total expectations.

Next, we decompose the regret into two parts: one that occurs with high probability (event $\mathcal{E}_\tau^C$) and one that occurs with low probability (event $\mathcal{E}_\tau$):

$$
\mathbb{E}[\Delta R_\tau] = \mathbb{E}\left[\Delta R_\tau \mathbb{I}(\mathcal{E}_{\tau-1}) + \Delta R_\tau \mathbb{I}(\mathcal{E}_{\tau-1}^C)\right].
$$

We bound the contribution of the low-probability event by $(N+1)\mathbb{P}(\mathcal{E}\tau - 1)$, which is small due to our concentration results.

For the high-probability event, we show that the difference between the optimistic and true revenues is bounded by $\Delta_{i,\tau}$. Applying Lemma B.1, we can then bound the regret for each epoch:

$$
\mathbb{E}[\Delta R_\tau] \leq (N+1)\mathbb{P}(\mathcal{E}\tau - 1) + \mathbb{E}\left[\left(1 + \psi_\mu v_{0d} + \sum_{j \in S_\tau} \psi_\mu v_j\right)(R(S_\tau; \bar{v}_\tau) - R(S_\tau; \psi_\mu v))\mathbb{I}(\mathcal{E}_{\tau-1}^C)\right].
$$

Summing over all epochs and using the properties of the epoch duration, we show that the total regret is bounded by $\tilde{O}(\sqrt{NT})$. $\square$

The full detailed proof is provided in Appendix B.

Next, we provide a lower bound result for the regret in the following theorem:

**Theorem 5.2** *For any policy $\pi$, suppose $K \leq N/4$, $T \geq 1$, and $\psi_\mu \in (0, 1)$. There exists a universal constant $c > 0$ such that*

$$Reg(T, \pi) \geq c \min \left\{ T, \sqrt{\frac{TN}{\psi_\mu}} \right\}.$$

The proof of Theorem 5.2 is deferred to Appendix C. This theorem establishes a lower bound on the regret, showing that no policy can achieve a better regret rate than $\Omega(\sqrt{NT})$.

**Remark 5.3** *The effect of the threshold $\mu$ in the upper bound appears only in logarithmic terms, suggesting that the regret increases with larger $\mu$. In contrast, in the lower bound, $\psi_\mu$ appears in the square root and the denominator, indicating that the regret decreases with larger $\mu$. It is important to note that $\mu$ is determined by business conditions and is typically fixed. We conjecture that the upper bound is not tight concerning $\mu$, indicating potential areas for future improvement in the analysis.*

## 6 Non-Thresholded Setting: The Patient Learner

In this section, we modify our algorithm for environments that do not apply a threshold for delays. We refer to the seller in this setting as the *patient* learner. The patient learner is assumed to have knowledge of the expected delay. This assumption is consistent with existing literature (see e.g. Joulani et al. [2013], Blanchet et al. [2024]).

We build our concentration result as follows:

**Lemma 6.1** *With probability at least $1 - O(N^{-2}T^{-1})$ we have*

$$|\hat{v}_{i,\tau} - v_i| \leq \sqrt{\frac{48\hat{v}_{i,\tau} \log(NT)}{|E_i(\tau)|}} + \frac{48 \log(NT)}{|E_i(\tau)|} + \frac{\mathbb{E}[d_s]}{|E_i(\tau)|} + \frac{\sqrt{6\mathbb{E}[d_s] \log(NT)}}{|E_i(\tau)|}.$$

The proof is deferred to the Appendix. According to Lemma 6.1, we modify Algorithm 1 by changing $\Delta_{i,\tau}$ to

$$\tilde{\Delta}_{i,\tau} = \sqrt{\frac{48\hat{v}_{i,\tau} \log(NT)}{|E_i(\tau)|}} + \frac{48 \log(NT)}{|E_i(\tau)|} + \frac{\mathbb{E}[d_s]}{|E_i(\tau)|} + \frac{\sqrt{6\mathbb{E}[d_s] \log(NT)}}{|E_i(\tau)|}.$$

We then state the regret result of the modified algorithm:

**Theorem 6.2** *Let $\pi^{PA-DEMBA}$ be the policy produced by Algorithm 1 using $\tilde{\Delta}_{i,\tau}$. $\pi^{PA-DEMBA}$ satisfies*

$$Reg(T, \pi^{PA-DEMBA}) \leq \log(T) + K\sqrt{48T \log(NT)}$$
$$+ (K+1)(48 + \mathbb{E}[d_s] + \sqrt{6\mathbb{E}[d_s]}) \log^2(NT) + 72\sqrt{NT \log(NT)}.$$

**Remark 6.3** *In the non-thresholded setting, the term with $\psi_\mu$ disappears since $\lim_{\mu \to \infty} \psi_\mu = 1$. This implies that the regret in this setting does not depend on $\psi_\mu$, providing a potentially tighter bound compared to the thresholded case. However, new terms involving $\mathbb{E}[d_s]$ are introduced. Asymptotically, both the thresholded and non-thresholded regret bounds simplify to $\tilde{O}(\sqrt{NT})$, with the differences primarily reflected in the constants, and both bounds approach the lower bound.*

**Remark 6.4** *Incorporating the skewness or variance of the delay distribution can improve the regret upper bound in practice, particularly in the non-thresholded setting. For instance, distributions with faster decay rates, such as Gaussian distributions, may lead to better regret performance compared to long-tail distributions. This would involve using techniques like Bernstein-type inequalities or making assumptions about tail behavior (e.g., sub-exponential tails). However, in the current analysis, we focus on the expectation of the delay, and further improvements based on skewness are left for future work. The asymptotic regret bound remains $O(\sqrt{NT})$, independent of skewness.*

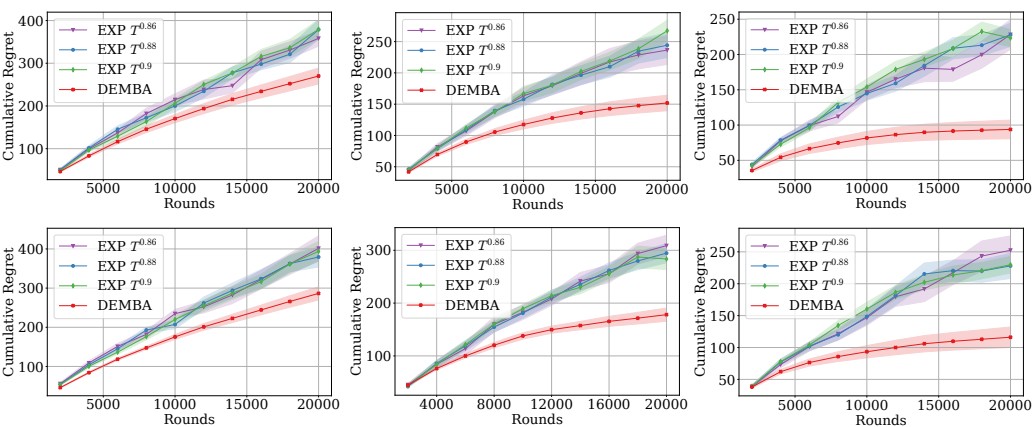

Figure 1: Simulation results of DEMBA algorithm with benchmarks. Top row: geometric delays. Bottom row: uniform delays. Left: $\mathbb{E}[d_s] = 500$, $\mu = 100$; Middle $\mathbb{E}[d_s] = 100$, $\mu = 100$; Right: $\mathbb{E}[d_s] = 100$, $\mu = 500$. Results are averaged over 100 independent runs.

# 7 Experiments

We conducted two sets of experiments to evaluate the performance of our algorithms. Our benchmark is an explore-then-exploit (EXP) algorithm, which explores by offering random assortments until a pre-specified time and then offers the optimized assortment based on the collected data. We tuned the exploration duration of the EXP algorithm and used the three best-performing durations in our comparisons.

We used $N = 10$, $K = 4$ and $p_i = 1$ for all $i \in \{1, \ldots, N\}$. The attraction parameters were set as:

$$v_i = \begin{cases} 0.25 + \epsilon & if \quad i \in \{1, 2, 9, 10\} \\ 0.25 & otherwise, \end{cases}$$

where $\epsilon$ represents the contrast between products. With this setting the optimal assortment is $\{1, 2, 9, 10\}$.

In the first set of experiments, we tested our algorithm in two different delay settings: geometrically distributed and uniformly distributed delays. We set $\epsilon = 0.05$ and used three cases in each distribution with increasing $\psi_\mu$ values: $\mathbb{E}[d_s] = 500$, $\mu = 100$; $\mathbb{E}[d_s] = 100$, $\mu = 100$; and $\mathbb{E}[d_s] = 100$, $\mu = 500$. The results of this experiment are shown in Figure 1. We observed that the DEMBA algorithm learns effectively and performs better than our benchmarks in all settings. Learning becomes more difficult as $\psi_\mu$ decreases due to increased censorship and information loss from thresholding. With uniform delays, the learning is more challenging due to the heavy tail of the distribution. The gaps between DEMBA and the benchmarks increase with higher $\psi_\mu$, suggesting better utilization of information by DEMBA.

In our second set of experiments, we tested how the contrast parameter $\epsilon$ affects learning and how the PA-DEMBA algorithm performs. We used geometric delays with $\mathbb{E}[d_s] = 100$ and $\mu = 100$ for the first experiment and $\mathbb{E}[d_s] = 100$ for the second experiment. The results are shown in Figure 2. On the left-hand side, we observe that when the number of rounds is low (and thus the amount of learning is limited), lower contrast values lead to better results. As the number of rounds (and therefore the amount of learning) increases, as expected, higher contrast simplifies the learning problem, as indicated by lower regret curves. Furthermore, on ther right hand side, the PA-DEMBA algorithm demonstrated robust performance, effectively managing the challenges posed by the non-thresholded setting.

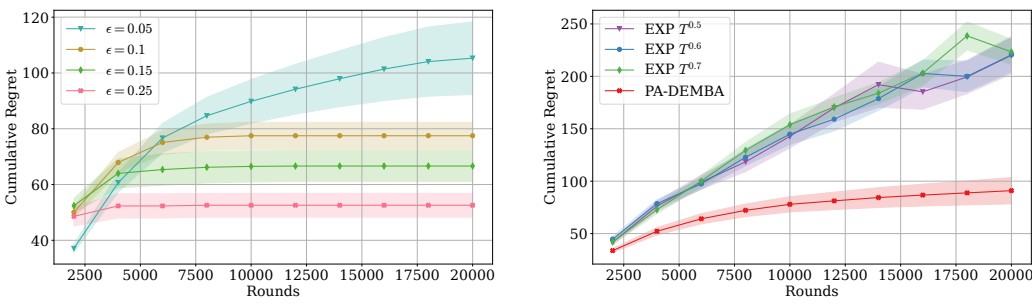

Figure 2: Left: Performance change of DEMBA algorithm with different contrast levels. Right: PA-DEMBA and benchmarks with non-thresholded delays. Results are averaged over 100 independent runs.

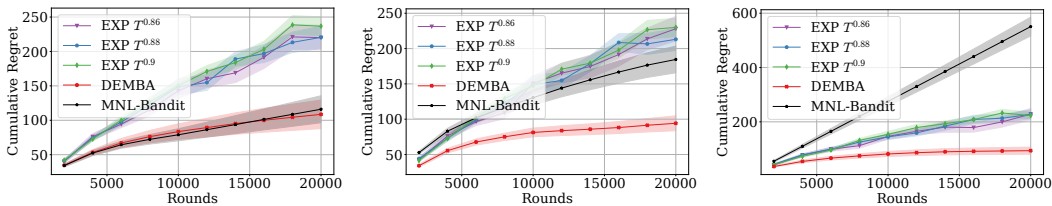

Figure 3: Comparison with MNL-Bandit. Left: no delay; Middle $\mathbb{E}[d_s] = 50$; Right: $\mathbb{E}[d_s] = 100$. Results are averaged over 100 independent runs.

In our third set of experiments, we compare DEMBA algorithm and EXP benchmarks with MNL-Bandit algorihm from Agrawal et al. [2019]. While MNL-Bandit learns customer preferences similarly to DEMBA, it does not account for potential delays in the feedback. In this experiment, we considered $\mu = 500$ and we applied a geometric delay distribution with no delay, $\mathbb{E}[d_s] = 50$ and $\mathbb{E}[d_s] = 100$. We observed that when there is no delay, the performance of MNL-Bandit and DEMBA is almost identical. However, as the delay increases, the performance of MNL-Bandit deteriorates, clearly indicating that it fails to handle delayed feedback.

# 8   Conclusion

This work provides the first solution and analysis for delayed choice modeling and online assortment optimization. We introduced two novel algorithms, DEMBA for thresholded feedback settings and PA-DEMBA for non-thresholded settings, demonstrating their effectiveness through theoretical guarantees and comprehensive experiments. Our algorithms address the dual challenges of unknown Multinomial Logit (MNL) parameters and delayed feedback, achieving sub-linear regret bounds.

Lastly, we discuss future work. Our lower bound suggest an improvement on regret by considering the delay distribution via $\psi_\mu$. This would require learning the delay distribution itself, adding more complexity. Moreover, it could be interesting to explore scenarios where no-purchase decisions are indistinguishable from delayed purchases, such as settings where tracking no purchases is not possible. Additionally, it would be worthwhile to consider multi-level choice settings where customer preferences and rewards are revealed to the seller in multiple stages with delays between each stage.

## Acknowledgments

I would like to thank the anonymous referees for their valuable feedback, which has helped to improve this paper.

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

# A  Proof of Lemma 4.1

We begin by providing an instrumental theorem that will be useful to establish concentration results.

**Theorem A.1 (Theorem 5 of Agrawal et al. [2019])** *Consider $n$ i.i.d. geometric random variables $X_1, \ldots, X_n$ with parameter $p$, i.e. for any $i$*

$$\mathbb{P}(X_i = m) = (1-p)^m p \quad \forall m = \{0, 1, 2, \ldots\},$$

*and let $\mu = \mathbb{E}(X_i) = \frac{1-p}{p} \le 1$ and $\bar{X} = \frac{1}{n} \sum_{i=1}^n X_i$. For any $\zeta > 0$, we have*

$$\mathbb{P}(\bar{X} > (1+\zeta)\mu) \le \exp\left(-\frac{n\mu\zeta^2}{2(1+\zeta)(1+\mu)^2}\right).$$

We now provide the proof of Lemma 4.1 that is crucial for our subsequent analysis.

*Proof of Lemma 4.1.* Note that $\hat{v}_{i,\tau}$ is a biased estimator and defined as

$$\hat{v}_{i,\tau} = \frac{\tilde{v}_{i,\tau}}{|E_\tau(i)|} = \frac{\sum_{s=1}^{t_\tau^{end}} o_{i,s,t_\tau^{end}}}{|E_\tau(i)|},$$

where $|E_\tau(i)|$ is the total number of epochs that product $i$ is shown to the customer and $\sum_{s=1}^{t_\tau^{end}} o_{i,s,t_\tau^{end}}$ is the total observed sales of product $i \ne 0d$ or is the total observed delayed no selections.

Let $t$ be the current period, i.e. $t = t_\tau^{end} + 1$. We have

$$
\begin{aligned}
|\hat{v}_{i,\tau} - \psi_\mu v_i| &= \left| \frac{\sum_{s=1}^{t-1} o_{i,s,t-1}}{|E_\tau(i)|} - \psi_\mu v_i \right| \\
&= \left| \frac{\sum_{s=1}^{t-\mu-1} a_{i,s}\mathbb{I}(d_s \le \mu) + \sum_{s=t-\mu}^{t-1} a_{i,s}\mathbb{I}(d_s \le t-s)}{|E_\tau(i)|} - \psi_\mu v_i \right| \\
&= \left| \frac{\sum_{s=1}^{t-1} a_{i,s}\mathbb{I}(d_s \le \mu) + \sum_{s=t-\mu}^{t-1} a_{i,s}\big(\mathbb{I}(d_s \le t-s) - \mathbb{I}(d_s \le \mu)\big)}{|E_\tau(i)|} - \psi_\mu v_i \right| \\
&\le \left| \frac{\sum_{s=1}^{t-1} a_{i,s}\mathbb{I}(d_s \le \mu)}{|E_\tau(i)|} - \psi_\mu v_i \right| + \left| \frac{\sum_{s=t-\mu}^{t-1} a_{i,s}\big(1 - \mathbb{I}(d_s \le \mu)\big)}{|E_\tau(i)|} \right|,
\end{aligned}
\tag{5}
$$

where the second equality follows from the decomposition of old observations before the threshold and recent observations, the third equality is rearrangement of the values and the last inequality follows from triangle inequality.

For the first element of (5), we have

$$
\begin{aligned}
\left| \frac{\sum_{s=1}^{t-1} a_{i,s}\mathbb{I}(d_s \le \mu)}{|E_\tau(i)|} - \psi_\mu v_i \right| &= \left| \frac{\sum_{s=1}^{t-1} a_{i,s}\mathbb{I}(d_s \le \mu)}{|E_\tau(i)|} - \frac{\psi_\mu \sum_{s=1}^{t-1} a_{i,s}}{|E_\tau(i)|} + \frac{\psi_\mu \sum_{s=1}^{t-1} a_{i,s}}{|E_\tau(i)|} - \psi_\mu v_i \right| \\
&\le \left| \frac{\sum_{s=1}^{t-1} a_{i,s}\mathbb{I}(d_s \le \mu)}{|E_\tau(i)|} - \frac{\psi_\mu \sum_{s=1}^{t-1} a_{i,s}}{|E_\tau(i)|} \right| + \left| \frac{\psi_\mu \sum_{s=1}^{t-1} a_{i,s}}{|E_\tau(i)|} - \psi_\mu v_i \right| \\
&\le \underbrace{\left| \frac{\sum_{s=1}^{t-1} \mathbb{I}(d_s \le \mu)}{|E_\tau(i)|} - \psi_\mu \right|}_{(a)} + \underbrace{\left| \frac{\sum_{s=1}^{t-1} a_{i,s}}{|E_\tau(i)|} - v_i \right|}_{(b)}
\end{aligned}
\tag{6}
$$

We have

$$(a) \le \sqrt{\frac{\log(NT)}{|E_\tau(i)|}}, \tag{7}$$

by Hoeffding's inequality.

For part $(b)$, we make use of the Chernoff bound from Theorem A.1 in two cases according to $\zeta = (v_i + 1)\sqrt{\frac{6\log(NT)}{v_i|E_\tau(i)|}}$.

*Case 1 $\zeta \leq \frac{1}{2}$:* We have from Theorem A.1

$$\mathbb{P}\left(\frac{\sum_{s=1}^{t-1} a_{i,s}}{|E_\tau(i)|} - v_i > \zeta v_i\right) \leq \frac{1}{N^2 T^2},$$

$$\mathbb{P}\left(\frac{\sum_{s=1}^{t-1} a_{i,s}}{|E_\tau(i)|} - v_i < -\zeta v_i\right) \leq \frac{1}{N^2 T^2},$$

$$\mathbb{P}\left(\left|\frac{\sum_{s=1}^{t-1} a_{i,s}}{|E_\tau(i)|} - v_i\right| > (v_i + 1)\sqrt{\frac{6v_i\log(NT)}{|E_\tau(i)|}}\right) \leq \frac{2}{N^2 T^2}.$$

Therefore, we have

$$\mathbb{P}\left(\left|\frac{\sum_{s=1}^{t-1} a_{i,s}}{|E_\tau(i)|} - v_i\right| > \sqrt{\frac{24v_i\log(NT)}{|E_\tau(i)|}}\right) \leq \frac{2}{N^2 T^2}. \tag{8}$$

We shall generalize the result at 8 in two cases for substituting $v_i$ with $\hat{v}_{i,\tau}$ in the concentration radius and upper bounding the new bound utilizing $v_i$.

For $\hat{v}_{i,\tau}$, we have

$$\mathbb{P}(\hat{v}_{i,\tau} - v_i < -v_i\frac{1}{2}) \leq \frac{1}{N^2 T^2},$$

hence,

$$\mathbb{P}(2\hat{v}_{i,\tau} \leq v_i) \leq \frac{1}{N^2 T^2}.$$

Combining this with 8, we get

$$\mathbb{P}\left(\left|\frac{\sum_{s=1}^{t-1} a_{i,s}}{|E_\tau(i)|} - v_i\right| > \sqrt{\frac{48\hat{v}_{i,\tau}\log(NT)}{|E_\tau(i)|}}\right) \leq \frac{3}{N^2 T^2}. \tag{9}$$

For upper bounding 9 using $v_i$, we have

$$\mathbb{P}(\hat{v}_{i,\tau} - v_i > v_i\frac{1}{2}) \leq \frac{1}{N^2 T^2},$$

hence

$$\mathbb{P}(\frac{3v_i}{2} \leq \hat{v}_{i,\tau}) \leq \frac{1}{N^2 T^2}.$$

Therefore, we conclude

$$\mathbb{P}\left(\left|\frac{\sum_{s=1}^{t-1} a_{i,s}}{|E_\tau(i)|} - v_i\right| > \sqrt{\frac{72v_i\log(NT)}{|E_\tau(i)|}}\right) \leq \frac{3}{N^2 T^2}. \tag{10}$$

*Case 2 $\zeta > \frac{1}{2}$:* We have $2\zeta^2 \geq \frac{1}{2}$. Let $\zeta' = 2\zeta^2$. We have by Theorem A.1

$$\mathbb{P}\left(\left|\frac{\sum_{s=1}^{t-1} a_{i,s}}{|E_\tau(i)|} - v_i\right| > \zeta' v_i\right) \leq \exp\left(-\frac{|E_\tau(i)|v_i\zeta'^2}{2(1+\zeta')(1+v_i)^2}\right)$$

$$\leq \exp\left(-\frac{|E_\tau(i)|v_i\zeta'}{6(1+v_i)^2}\right),$$

substituting the value of $\zeta'$, we get

$$\mathbb{P}\left(\left|\frac{\sum_{s=1}^{t-1} a_{i,s}}{|E_\tau(i)|} - v_i\right| > \frac{48 \log(NT)}{|E_\tau(i)|}\right) \le \frac{1}{N^2 T^2}. \tag{11}$$

Combining 9 and 11, and applying union bound we have

$$\mathbb{P}\left(\left|\frac{\sum_{s=1}^{t-1} a_{i,s}}{|E_\tau(i)|} - v_i\right| \le \sqrt{\frac{48 \hat{v}_{i,\tau} \log(NT)}{|E_\tau(i)|}} + \frac{48 \log(NT)}{|E_\tau(i)|}\right) \ge 1 - \frac{4}{N^2 T}.$$

Additionally, combining 9, 10 and 11, and applying union bound we have

$$\mathbb{P}\left(\left|\frac{\sum_{s=1}^{t-1} a_{i,s}}{|E_\tau(i)|} - v_i\right| > \sqrt{\frac{72 v_i \log(NT)}{|E_\tau(i)|}} + \frac{48 \log(NT)}{|E_\tau(i)|}\right) \tag{12}$$

$$\ge \mathbb{P}\left(\left|\frac{\sum_{s=1}^{t-1} a_{i,s}}{|E_\tau(i)|} - v_i\right| \le \sqrt{\frac{48 \hat{v}_{i,\tau} \log(NT)}{|E_\tau(i)|}} + \frac{48 \log(NT)}{|E_\tau(i)|}\right) \ge 1 - \frac{4}{N^2 T}. \tag{13}$$

We can bound the second element of (1) as

$$\left|\frac{\sum_{s=t-\mu}^{t-1} a_{i,s}\left(1 - \mathbb{I}(d_s \le \mu)\right)}{|E_\tau(i)|}\right| \le \frac{\mu}{|E_\tau(i)|}. \tag{14}$$

Combining (7), (12) and (14) gives the result.

$\square$

## B  Proof of Theorem 5.1

We begin with establishing a key result that will be instrumental in our main proof.

**Lemma B.1** *Let $S^*$ represent the optimal assortment when the MNL parameters are given by $v$. Also, let $S_\tau$ denote the assortment applied by the policy at epoch $\tau$. For any parameter set $w$ distinct from $v$ the following hold.*

*i. If $v_i \le w_i$ for all $i \in 1, \ldots, N$, and $v_{0d} \ge w_{0d}$, then $R(S^*; w) \ge R(S^*; v)$.*

*ii. If $v_i \le w_i$ for all $i \in 1, \ldots, N$, and $v_{0d} = w_{0d}$, then $R(S_\tau; w) - R(S_\tau; v) \le \frac{\sum_{i \in S_e}(w_i - v_i)}{1 + v_{0d} + \sum_{j \in S_e} v_j}$.*

*Proof of Lemma B.1.* Note that the proof of *Part i.* mostly resembles Lemma A.3 in Agrawal et al. [2019], and we write the whole proof here for being self-contained.

*Proof of Part i.* Let $v^j$ be identical to $v$ except for the $j^{th}$ element, which is increased to $w_j$. We aim to show that for any $j \in S^*$, if $v_j$ is increased to $w_j$, then $R(S^*; v^j) \ge R(S^*; v)$. This suffices to prove $R(S^*; w) \ge R(S^*; v)$.

Define $T := \sum_{i \in S^* \setminus j} r_i v_i$ and $V := 1 + \sum_{i \in S^* \setminus j} v_i$. If there exists a $j \in S^*$ such that $r_j < R(S^*)$, removing product $j$ from the assortment yields higher expected revenue, contradicting the optimality of $S^*$. Therefore, we have

$$\begin{aligned}
r_j &\ge R(S^*; v) \\
&= \frac{\sum_{i \in S^*} r_i v_i}{1 + \sum_{i \in S^*} v_i} \\
&= \frac{\sum_{i \in S^* \setminus j} r_i v_i + r_j v_j}{1 + \sum_{i \in S^* \setminus j} v_i + v_j} \\
&= \frac{T + r_j v_j}{V + v_j}.
\end{aligned}$$

for all $j \in S^*$. Rearranging terms, we get

$$r_j V \geq T. \tag{15}$$

To have $R(S^*; v^j) \geq R(S^*; v)$, we need to show that

$$\frac{T + r_j w_j}{V + w_j} \geq \frac{T + r_j v_j}{V + v_j},$$

which is equivalent to

$$Tv_j + r_j V w_j \geq T w_j + r_j V v_j.$$

Rearranging terms, we get

$$r_j V(w_j - v_j) \geq T(w_j - v_j),$$

which holds thanks to (15). Moreover, the case for $i = 0d$ holds trivially which concludes our proof.

*Proof of Part ii.* We have

$$
\begin{aligned}
R(S_\tau; w) - R(S_\tau; v) &= \sum_{i \in S_e} r_i \frac{w_i}{1 + v_{0d} + \sum_{j \in S_e} w_j} - \sum_{i \in S_e} r_i \frac{v_i}{1 + v_{0d} + \sum_{j \in S_e} v_j} \\
&\leq \sum_{i \in S_e} r_i \frac{w_i}{1 + v_{0d} + \sum_{j \in S_e} w_j} - \sum_{i \in S_e} r_i \frac{v_i}{1 + v_{0d} + \sum_{j \in S_e} w_j} \\
&= \frac{\sum_{i \in S_e} r_i (w_i - v_i)}{\left(1 + v_{0d} + \sum_{j \in S_e} w_j\right)} \\
&\leq \frac{\sum_{i \in S_e} (w_i - v_i)}{1 + v_{0d} + \sum_{j \in S_e} v_j},
\end{aligned}
$$

where the second inequality follows from $r_i \leq 1, \forall i \in [N]$.

$\square$

Now, we provide the proof of Theorem 5.1.

*Proof of Theorem 5.1.* We have

$$R(S_\tau; \bar{v}_\tau) \geq R(S^*; \bar{v}_\tau) \geq R(S^*; \psi_\mu v), \tag{16}$$

where the first inequality holds thanks to the definition of $S_\tau$ and the second inequality follows from Lemma B.1 with probability at least $1 - O(N^{-1} T^{-1})$.

We have

$$\text{Reg}(T, \pi) = \mathbb{E}\left[ \sum_{\tau=1}^{\bar{\tau}} |\mathcal{H}_\tau| \left( R(S^*; \psi_\mu v) - R(S_\tau; \psi_\mu v) \right) \right],$$

where $\mathcal{H}_\tau$ is the duration of epoch $\tau$ and $|\mathcal{H}_\tau| \sim Geom(\frac{1}{1 + \psi_\mu v_{0d} + \sum_{j \in S_\tau} \psi_\mu v_j})$, then $\mathbb{E}[|\mathcal{H}_\tau| \mid S_\tau] = 1 + \psi_\mu v_{0d} + \sum_{j \in S_\tau} \psi_\mu v_j$. Since $S_\tau$ is determined by the history of policy($\mathcal{F}_{\tau-1}$), we have $\mathbb{E}[|\mathcal{H}_\tau| \mid \mathcal{F}_{\tau-1}] = 1 + \psi_\mu v_{0d} + \sum_{j \in S_\tau} \psi_\mu v_j$. Therefore the regret is equal to

$$\text{Reg}(T, \pi) = \mathbb{E}\left[ \sum_{\tau=1}^{\bar{\tau}} \mathbb{E}\left[ |\mathcal{H}_\tau| \left( R(S^*; \psi_\mu v) - R(S_\tau; \psi_\mu v) \right) \mid \mathcal{F}_{\tau-1} \right] \right],$$

and by the law of total expectations

$$\text{Reg}(T, \pi) = \mathbb{E}\left[ \sum_{\tau=1}^{\bar{\tau}} \left( 1 + \psi_\mu v_{0d} + \sum_{j \in S_\tau} \psi_\mu v_j \right) \left( R(S^*; \psi_\mu v) - R(S_\tau; \psi_\mu v) \right) \right].$$

Let the epoch based regret be $\Delta R_\tau$, we can write it as

$$\Delta R_\tau = \left(1 + \psi_\mu v_{0d} + \sum_{j \in S_\tau} \psi_\mu v_j\right) \left(R(S^*; \psi_\mu v) - R(S_\tau; \psi_\mu v)\right),$$

then the regret can be written in the form of

$$\text{Reg}(T, \pi) = \mathbb{E}\left[\sum_{\tau=1}^{\bar{\tau}} \Delta R_\tau\right].$$

We define event $\mathcal{E}$, for all $\tau$ as

$$\mathcal{E}_\tau := \bigcup_{i=1}^{N} \left\{|\bar{v}_{i,\tau} - \psi_\mu v_i| > \Delta_\tau\right\} \cup \left\{|\psi_\mu v_{0d} - \underline{v}_{0d,\tau}| > \Delta_\tau\right\}.$$

The event $\mathcal{E}_\tau$ is a low probability event. We will analyze the regret according to the event separately:

$$\mathbb{E}[\Delta R_\tau] = \mathbb{E}\left[\Delta R_\tau \mathbb{I}(\mathcal{E}_{\tau-1}) + \Delta R_\tau \mathbb{I}(\mathcal{E}_{\tau-1}^C)\right].$$

We have $\Delta R_\tau \leq N + 1$ and hence

$$\mathbb{E}[\Delta R_\tau] \leq (N+1)\mathbb{P}(\mathcal{E}_{\tau-1}) + \mathbb{E}\left[\Delta R_\tau \mathbb{I}(\mathcal{E}_{\tau-1}^C)\right].$$

We have by (16) that

$$\mathbb{E}[\Delta R_\tau] \leq (N+1)\mathbb{P}(\mathcal{E}_{\tau-1}) + \mathbb{E}\left[\left(1 + \psi_\mu v_{0d} + \sum_{j \in S_\tau} \psi_\mu v_j\right)\left(R_\tau(S_\tau; \bar{v}_\tau) - R(S_\tau; \psi_\mu v)\right)\mathbb{I}(\mathcal{E}_{\tau-1}^C)\right].$$

Then, by Lemma B.1 we have

$$\left(1 + \psi_\mu v_{0d} + \sum_{j \in S_\tau} \psi_\mu v_j\right)\left(R(S_\tau; \bar{v}_\tau) - R(S_\tau; \psi_\mu v)\right)\mathbb{I}(\mathcal{E}_{\tau-1}^C)$$

$$\leq \left(1 + \psi_\mu v_{0d} + \sum_{j \in S_\tau} \psi_\mu v_j\right)\left(R(S_\tau; \bar{v}_\tau) - R(S_\tau; \psi_\mu v)\right)$$

$$= \left(1 + \psi_\mu v_{0d} + \sum_{j \in S_\tau} \psi_\mu v_j\right)\left(\frac{\sum_{i \in S_\tau} r_i \bar{v}_{i,\tau}}{1 + \underline{v}_{0d} + \sum_{j \in S_\tau} \bar{v}_j} - \frac{\sum_{i \in S_\tau} r_i \bar{v}_{i,\tau}}{1 + \psi_\mu v_{0d} + \sum_{j \in S_\tau} \bar{v}_j}\right.$$

$$\left. + \frac{\sum_{i \in S_\tau} r_i \bar{v}_{i,\tau}}{1 + \psi_\mu v_{0d} + \sum_{j \in S_\tau} \bar{v}_j} - \frac{\sum_{i \in S_\tau} r_i \psi_\mu v_i}{1 + \psi_\mu v_{0d} + \sum_{j \in S_\tau} \psi_\mu v_j}\right)$$

$$\leq \left(1 + \psi_\mu v_{0d} + \sum_{j \in S_\tau} \psi_\mu v_j\right)\left(\frac{\sum_{i \in S_\tau} r_i \bar{v}_{i,\tau}(\psi_\mu v_{0d} - \underline{v}_{0d})}{\left(1 + \underline{v}_{0d} + \sum_{j \in S_\tau} \bar{v}_j\right)\left(1 + \psi_\mu v_{0d} + \sum_{j \in S_\tau} \bar{v}_j\right)} + \frac{\sum_{i \in S_\tau} r_i(\bar{v}_{i,\tau} - \psi_\mu v_i)}{1 + \psi_\mu v_{0d} + \sum_{j \in S_\tau} \bar{v}_j}\right)$$

$$\leq \left(1 + \psi_\mu v_{0d} + \sum_{j \in S_\tau} \psi_\mu v_j\right)\left(\frac{K(\psi_\mu v_{0d} - \underline{v}_{0d})}{\left(1 + \underline{v}_{0d} + \sum_{j \in S_\tau} \bar{v}_j\right)\left(1 + \psi_\mu v_{0d} + \sum_{j \in S_\tau} \bar{v}_j\right)} + \frac{\sum_{i \in S_\tau}(\bar{v}_{i,\tau} - \psi_\mu v_i)}{1 + \psi_\mu v_{0d} + \sum_{j \in S_\tau} \bar{v}_j}\right)$$

$$\leq \left(1 + \psi_\mu v_{0d} + \sum_{j \in S_\tau} \psi_\mu v_j\right)\left(\frac{K(\psi_\mu v_{0d} - \underline{v}_{0d})}{1 + \psi_\mu v_{0d} + \sum_{j \in S_\tau} \psi_\mu v_j} + \frac{\sum_{i \in S_\tau}(\bar{v}_{i,\tau} - \psi_\mu v_i)}{1 + \psi_\mu v_{0d} + \sum_{j \in S_\tau} \psi_\mu v_j}\right)$$

$$= K(\psi_\mu v_{0d} - \underline{v}_{0d}) + \sum_{i \in S_\tau}(\bar{v}_{i,\tau} - \psi_\mu v_{i,\tau}).$$

Let $E_i$ be the number of periods that the product $i$ is offered. Then, the regret can be written as

$$\text{Reg}(T, \pi) \leq \mathbb{E}\left[\sum_{\tau=1}^{\bar{\tau}}\left((N+1)\mathbb{P}(\mathcal{E}_{\tau-1}) + K\Delta_{0d,\tau} + \sum_{i \in S_\tau}\Delta_{i,\tau}\right)\right]$$

$$\leq \mathbb{E}\left[\sum_{\tau=1}^{\bar{\tau}}\left(\frac{N+1}{NT^2} + K\sqrt{\frac{(72v_{0d}+1)\log(NT)}{\tau}} + \frac{48K(\log(NT)+\mu)}{\tau}\right.\right.$$

$$\left.\left. + \sum_{i \in S_\tau}\left(\sqrt{\frac{(72v_i+1)\log(NT)}{|E_\tau(i)|}} + \frac{48\log(NT)+\mu}{|E_\tau(i)|}\right)\right)\right]$$

$$\leq \log(T) + K\sqrt{73T\log(NT)} + 48K\log(T)(\log(NT)+\mu) + 73\mathbb{E}\left[\sum_{i=1}^{N}\sqrt{v_i E_i \log(NT)}\right]$$

$$+ 48\log^2(NT) + \mu\log(T)$$

$$\leq \log(T) + K\sqrt{73T\log(NT)} + 48K\log(T)(\log(NT)+\mu) + 73\sum_{i=1}^{N}\sqrt{v_i\mathbb{E}[E_i]\log(NT)}$$

$$+ 48\log^2(NT) + \mu\log(T),$$

where the third inequality holds due to $\bar{\tau} \leq T$, $E_i \leq T$ and the fourth inequality holds due to Jensen's inequality.

Since, $\sum_{i=1}^{N} v_i\mathbb{E}[E_i] \leq T$, we conclude

$$\text{Reg}(T, \pi) \leq \log(T) + K\sqrt{73T\log(NT)} + 48K\log(T)(\log(NT)+\mu)$$
$$+ 73\sqrt{NT\log(NT)} + 48\log^2(NT) + \mu\log(T).$$

$\square$

## C    Proof of Theorem 5.2

Before giving the proof of Theorem 5.2, we first give an instrumental Lemma in the main proof and its proof.

**Lemma C.1** *Let $P$ and $Q$ be categorical distributions with $\mathbb{P}_P(X = i) = p_i$ and $\mathbb{P}_Q(X = i) = q_i$ where $p_i = q_i + \epsilon_i$ for all $i = 1, \ldots, I$. Also, let the selected option be observed with bias $\psi_\mu$. The Kullback-Leibler divergence between $P$ and $Q$, denoted by $D_{KL}(P \parallel Q)$, is given by*

$$D_{KL}(P \parallel Q) = \sum_{i=0}^{I}\frac{\psi_\mu\epsilon_i^2}{q_i}.$$

*Proof of Lemma C.1.* We have

$$D_{KL}(P \parallel Q) = \sum_{i=0}^{I}\psi_\mu(q_i + \epsilon_i)\log\left(\frac{\psi_\mu(q_i + \epsilon_i)}{\psi_\mu q_i}\right)$$

$$\leq \sum_{i=0}^{I}\psi_\mu(q_i + \epsilon_i)\frac{\epsilon_i}{q_i}$$

$$= \sum_{i=0}^{I}\frac{\psi_\mu\epsilon_i^2}{q_i},$$

where the inequality follows since $\log(1 + x) \leq x$ for any $x > -1$ and the last equality holds since $\sum_{i=0}^{I}\epsilon_i = 0$.

$\square$

*Proof of Theorem 5.2.* We set $p_i = 1$ for all $i \in [N]$. Let $S$ be a subset of $N$ with $|S| = K$. For attraction parameters, we set $v_0 = 1$, $v_{0d}$ arbitrary in $[0, 1]$, $v_i = (1 + \epsilon)/K$ for $i \in S$ and $v_i = 1/K$ otherwise.

We denote all subsets of product set $[N]$ of size $K$ as $\mathcal{S}_K$. We have

$$\max_{S \in \mathcal{S}_K} R(S) = R(S^*) = \frac{1 + \epsilon}{2 + v_{0d} + \epsilon}.$$

We employ the neighboring argument for assortment sets as described in Chen and Wang [2018]. For any arbitrary assortment $S_t$, we have

$$R(S_t) = \frac{1 + (1 - \delta)\epsilon}{2 + v_{0d} + (1 - \delta)\epsilon},$$

where $\delta$ is a measure of discrepancy from the optimal set $S$, i.e. $\delta := 1 - \frac{|S_t \cap S|}{K}$.

Then, we have

$$
\begin{aligned}
R(S^*) - R(S_t) &= \frac{1 + \epsilon}{2 + v_{0d} + \epsilon} - \frac{1 + (1 - \delta)\epsilon}{2 + v_{0d} + (1 - \delta)\epsilon} \\
&= \frac{\delta\epsilon + \delta\epsilon v_{0d}}{(2 + v_{0d} + \epsilon)(2 + v_{0d} + (1 - \delta)\epsilon)} \\
&\geq \frac{\delta\epsilon (1 + v_{0d})}{16}.
\end{aligned}
$$

For any arbitrary subset $S_t$ of $|S_t| \leq K$, we can find $\tilde{S}_t \in \mathcal{S}_K$ such that $S_t \subseteq \tilde{S}_t$. We define $\tilde{N}_i := \sum_{t=1}^{T} 1(i \in \tilde{S}_t)$. We have

$$\max_{S \in \mathcal{S}_K} \sum_{t=1}^{T} \mathbb{E}_S \left[ R(S) - R(S_t) \right] \tag{17}$$

$$\geq \max_{S \in \mathcal{S}_K} \sum_{t=1}^{T} \mathbb{E}_S \left[ R(S) - R(\tilde{S}_t) \right] \tag{18}$$

$$\geq \frac{1}{|\mathcal{S}_K|} \sum_{S \in \mathcal{S}_K} \sum_{t=1}^{T} \mathbb{E}_S \left[ R(S) - R(\tilde{S}_t) \right] \tag{19}$$

$$\geq \frac{1}{16|\mathcal{S}_K|} \sum_{S \in \mathcal{S}_K} \sum_{i \notin S} \mathbb{E}_S \left[ \tilde{N}_i \right] \frac{\epsilon}{K} \tag{20}$$

$$= \frac{\epsilon}{16} \left( T - \frac{1}{|\mathcal{S}_K|} \sum_{S \in \mathcal{S}_K} \sum_{i \in S} \frac{\mathbb{E}_S \left[ \tilde{N}_i \right]}{K} \right). \tag{21}$$

Let $\mathcal{S}_{K-1}^{(i)}$ be all subsets of $[N]$ of size $K - 1$ that do not include $i$. We have

$$
\begin{aligned}
\frac{1}{K|\mathcal{S}_K|} \sum_{S \in \mathcal{S}_K} \sum_{i \in S} \mathbb{E}_{S_1 \cup S_2^*} \left[ \tilde{N}_i \right] &= \frac{1}{K|\mathcal{S}_K|} \sum_{i=1}^{N} \sum_{S \in \mathcal{S}_K : i \in S} \mathbb{E}_S \left[ \tilde{N}_i \right] \\
&= \frac{1}{K|\mathcal{S}_K|} \sum_{i=1}^{N} \sum_{S' \in \mathcal{S}_{K-1}^{(i)}} \mathbb{E}_{S' \cup \{i\}} \left[ \tilde{N}_i \right].
\end{aligned}
\tag{22}
$$

We define probability measures $P$ and $Q$ through parameterizations $S'$ and $S' \cup \{i\}$ respectively. We denote Total Variation(TV) distance between two probability measures as $TV(P, Q) := \sup_A |P(A) - Q(A)|$ and Kullback-Leibler (KL) divergence as $D_{KL}(P \parallel Q)$. We have $0 \leq \tilde{N}_i \leq T$

and

$$|\mathbb{E}_P[\tilde{N}_i] - \mathbb{E}_Q[\tilde{N}_i]| \leq \sum_{j=0}^{T} j \left| P(\tilde{N}_i = j) - Q(\tilde{N}_i = j) \right|$$

$$\leq T \sum_{j=0}^{T} \left| P(\tilde{N}_i = j) - Q(\tilde{N}_i = j) \right|$$

$$\leq T \cdot TV(P, Q)$$

$$\leq T \cdot \sqrt{\frac{1}{2} D_{KL}(P \parallel Q)}, \tag{23}$$

where the last inequality follows from Pinsker's Inequality.

We use (23) in (22) and we get

$$\frac{1}{K|\mathcal{S}_K|} \sum_{S \in \mathcal{S}_K} \sum_{i \in S} \mathbb{E}_S \left[ \tilde{N}_i \right]$$

$$\leq \frac{1}{K|\mathcal{S}_K|} \sum_{i=1}^{N} \sum_{S' \in \mathcal{S}_{K-1}^{(i)}} \left( \mathbb{E}_{S'} \left[ \tilde{N}_i \right] + T \cdot \sqrt{\frac{1}{2} D_{KL}(P_{S'} \parallel P_{S' \cup \{i\}})} \right). \tag{24}$$

For the first element of (24), we have

$$\frac{1}{K|\mathcal{S}_K|} \sum_{i=1}^{N} \sum_{S' \in \mathcal{S}_{K-1}^{(i)}} \mathbb{E}_{S'} \left[ \tilde{N}_i \right] = \frac{1}{K|\mathcal{S}_K|} \sum_{S' \in \mathcal{S}_{K-1}} \sum_{i \notin S'} \mathbb{E}_{S'} \left[ \tilde{N}_i \right] \tag{25}$$

$$\leq \frac{1}{K|\mathcal{S}_K|} \sum_{S' \in \mathcal{S}_{K-1}} \sum_{i=1}^{N} \mathbb{E}_{S'} \left[ \tilde{N}_i \right] \tag{26}$$

$$= \frac{|\mathcal{S}_{K-1}|}{K|\mathcal{S}_K|} TK \tag{27}$$

$$= \frac{TK}{N - K + 1} \tag{28}$$

$$\leq \frac{T}{3}, \tag{29}$$

where the last inequality follows from $K \leq N/4$.

For the second element of (24), we have that

$$\frac{T}{K|\mathcal{S}_K|} \sum_{i=1}^{N} \sum_{S' \in \mathcal{S}_{K-1}^{(i)}} \sqrt{\frac{1}{2} D_{KL}(P_{S'} \parallel P_{S' \cup \{i\}})} \tag{30}$$

$$\leq \frac{T|\mathcal{S}_{K-1}|}{K|\mathcal{S}_K|} \sum_{i=1}^{N} \max_{S' \in \mathcal{S}_{K-1}^{(i)}} \sqrt{\frac{1}{2} D_{KL}(P_{S'} \parallel P_{S' \cup \{i\}})} \tag{31}$$

$$= \frac{T}{N - K + 1} \sum_{i=1}^{N} \max_{S' \in \mathcal{S}_{K-1}^{(i)}} \sqrt{\frac{1}{2} D_{KL}(P_{S'} \parallel P_{S' \cup \{i\}})} \tag{32}$$

$$\leq \max_{S' \in \mathcal{S}_{K-1}} T \sqrt{\frac{1}{2(N - K + 1)} \sum_{i \notin S'} D_{KL}(P_{S'} \parallel P_{S' \cup \{i\}})} \tag{33}$$

where the first inequality is due to Hölder's inequality and the second inequality follows from Jansen's inequality.

We have $D_{KL}(P_{S'}(\cdot|S_t) \parallel P_{S'\cup\{i\}}(\cdot|S_t)) = 0$ for $i \notin S_t$. For analyzing $i \in S_t$, we define $K' = |S_t| \leq K$, $J := |S_t \cap S'| \leq K - 1$ and $a = 1 + v_{0d} + \frac{K'}{K}$.

We have

$$|p_0 - q_0| = \left| \frac{1}{a + J\epsilon/K} - \frac{1}{a + (J+1)\epsilon/K} \right|$$

$$= \frac{\epsilon}{K(a^2 + aJ\epsilon/K + a(J+1)\epsilon/K + J(J+1)\epsilon^2/K^2)}$$

$$\leq \frac{\epsilon}{K};$$

$$|p_{0d} - q_{0d}| = v_{0d} \left| \frac{1}{a + J\epsilon/K} - \frac{1}{a + (J+1)\epsilon/K} \right|$$

$$= \frac{v_{0d}\epsilon}{K(a^2 + aJ\epsilon/K + a(J+1)\epsilon/K + J(J+1)\epsilon^2/K^2)}$$

$$\leq \frac{v_{0d}\epsilon}{K};$$

$$|p_j - q_j| \leq \frac{1+\epsilon}{K} \left| \frac{1}{a + J\epsilon/K} - \frac{1}{a + (J+1)\epsilon/K} \right|$$

$$= \frac{\epsilon + \epsilon^2}{K^2(a^2 + aJ\epsilon/K + a(J+1)\epsilon/K + J(J+1)\epsilon^2/K^2)}$$

$$\leq \frac{2\epsilon}{K^2} \quad \text{for } 1 \leq j \leq N \text{ and } j \neq i;$$

and

$$|p_i - q_i| \leq \frac{1}{K} \left| \frac{1}{a + J\epsilon/K} - \frac{1}{a + (J+1)\epsilon/K} \right| + \frac{\epsilon}{K} \frac{1}{a + J\epsilon/K}$$

$$\leq \frac{\epsilon}{K^2} + \frac{\epsilon}{K}$$

$$\leq \frac{2\epsilon}{K}.$$

We also have

$$q_0 = \frac{1}{a + (J+1)\epsilon/K} \geq \frac{1}{3 + \epsilon} \geq \frac{1}{3 + 1/2} \geq \frac{1}{4},$$

and

$$q_j = \frac{1+\epsilon}{K(a + (J+1)\epsilon/K)} \geq \frac{1+\epsilon}{4K} \geq \frac{1}{4K}.$$

Using these with Lemma C.1, we get

$$D_{KL}(P_{S'}(\cdot|S_t) \parallel P_{S'\cup\{i\}}(\cdot|S_t)) \leq \psi_\mu \left( \frac{4\epsilon^2}{K^2} + \frac{4v_{0d}^2\epsilon^2}{K^2} + \frac{4K \cdot J \cdot 4\epsilon^2}{K^4} + \frac{4K \cdot 4\epsilon^2}{K^2} \right)$$

$$\leq \psi_\mu \left( \frac{8\epsilon^2}{K^2} + \frac{16\epsilon_1^2}{K^2} + \frac{16\epsilon_1^2}{K} \right)$$

$$\leq \frac{40\psi_\mu\epsilon_1^2}{K}.$$

Integrating this into (33), we obtain

$$\max_{S' \in \mathcal{S}_{K-1}} T \sqrt{\frac{1}{2(N-K+1)} \sum_{i \notin S'} D_{KL}(P_{S'} \parallel P_{S' \cup \{i\}})} \tag{34}$$

$$\leq \max_{S' \in \mathcal{S}_{K-1}} T \sqrt{\frac{1}{2(N-K+1)} \sum_{i \notin S'} \mathbb{E}_{S'}[\tilde{N}_i] \frac{40\psi_\mu \epsilon^2}{K}} \tag{35}$$

$$\leq T \sqrt{\frac{1}{2(N-K+1)} \frac{40\psi_\mu \epsilon^2}{K} \sum_{i=1}^{N} \mathbb{E}_{S'}[\tilde{N}_i]} \tag{36}$$

$$\leq T \sqrt{\frac{1}{2(N-K+1)} \frac{40\psi_\mu \epsilon^2}{K} TK}. \tag{37}$$

For the final bound, we have

$$\max_{S \in \mathcal{S}_K} \sum_{t=1}^{T} \mathbb{E}_S \left[ R(S) - R(S_t) \right]$$

$$\geq \frac{\epsilon}{16} \left( T - \frac{1}{|\mathcal{S}_K|} \sum_{S \in \mathcal{S}_K} \sum_{i \in S} \frac{\mathbb{E}_S \left[ \tilde{N}_i \right]}{K} \right) \tag{38}$$

$$\geq \frac{\epsilon}{16} \left( T - \frac{T}{3} - T \sqrt{\frac{1}{2(N-K+1)} \frac{40\psi_\mu \epsilon^2}{K} TK} \right) \tag{39}$$

$$\tag{40}$$

where (38) follows from (21), (39) follows from (29) and (37).

Setting $\epsilon_1 = \min\{\sqrt{N/\psi_\mu T}, 0.5\}$ gives the final bound.

$\square$

# D  Proof of Theorem 6.2

We first provide an instrumental theorem that we will use to establish concentration results.

**Theorem D.1 (Theorem 4.4 of Mitzenmacher and Upfal [2017])** *Let $X_1, \ldots, X_n$ be independent Poisson trials such that $\mathbb{P}(X_i = 1) = p_i$. Let $X = \sum_{i=1}^{n} X_i$. Then, for $0 < \delta < 1$:*

$$\mathbb{P}(X \geq (1+\delta)\mathbb{E}[X]) \leq \exp(-\mathbb{E}[X]\delta^2/3).$$

*Proof of Lemma 6.1.*

We have

$$|\hat{v}_{i,\tau} - v_i| = \left| \frac{\sum_{s=1}^{t-1} o_{i,s,t-1}}{|E_i(\tau)|} - v_i \right|$$

$$= \left| \frac{\sum_{s=1}^{t-1} a_{i,s} \mathbb{I}(d_s \leq t - s)}{|E_i(\tau)|} - v_i \right|$$

$$= \left| \frac{\sum_{s=1}^{t-1} a_{i,s} - \sum_{s=1}^{t-1} a_{i,s} \mathbb{I}(d_s > t - s)}{|E_i(\tau)|} - v_i \right|$$

$$\leq \left| \frac{\sum_{s=1}^{t-1} a_{i,s}}{|E_i(\tau)|} - v_i \right| + \left| \frac{\sum_{s=1}^{t-1} \mathbb{I}(d_s > t - s)}{|E_i(\tau)|} \right|. \tag{41}$$

The derivation of the bound for the first element of (41) parallels that of Lemma 4.1, thus, we omit the analysis, yielding

$$\left| \frac{\sum_{s=1}^{t-1} a_{i,s}}{|E_i(\tau)|} - v_i \right| \leq \sqrt{\frac{48 \hat{v}_{i,\tau} \log(NT)}{|E_i(\tau)|}} + \frac{48 \log(NT)}{|E_i(\tau)|}, \tag{42}$$

with probability at least $1 - \frac{4}{N^2 T}$.

For the second element of (41), we first bound the expected number of unobserved feedback until time $t$. We have

$$\mathbb{E}\left[ \sum_{s=1}^{t-1} \mathbb{I}(d_s > t - s) \right] = \sum_{s=1}^{t-1} \mathbb{P}(d_s > t - s) = \sum_{n=1}^{t-1} \mathbb{P}(d_{t-n} > n)$$

$$\leq \sum_{n=0}^{\infty} \mathbb{P}(d_1 > n) = \sum_{n=0}^{\infty} \sum_{m=n+1}^{\infty} \mathbb{P}(d_1 = m)$$

$$= \sum_{m=0}^{\infty} \sum_{n=0}^{m-1} \mathbb{P}(d_1 = m) = \sum_{m=0}^{\infty} m \cdot \mathbb{P}(d_1 = m)$$

$$= \mathbb{E}[d_s]. \tag{43}$$

Then, we have

$$\left| \frac{\sum_{s=1}^{t-1} \mathbb{I}(d_s > t - s)}{|E_i(\tau)|} \right| = \frac{\sum_{s=1}^{t-1} \mathbb{I}(d_s > t - s)}{|E_i(\tau)|} \tag{44}$$

$$\leq \frac{(1 + \delta)\mathbb{E}\left[ \sum_{s=1}^{t-1} \mathbb{I}(d_s > t - s) \right]}{|E_i(\tau)|} \tag{45}$$

$$\leq \frac{(1 + \delta)\mathbb{E}[d_s]}{|E_i(\tau)|} \tag{46}$$

$$\leq \frac{\mathbb{E}[d_s]}{|E_i(\tau)|} + \frac{\sqrt{6\mathbb{E}[d_s] \log(NT)}}{|E_i(\tau)|} \tag{47}$$

where the first inequality follows by Theorem D.1 with probability at least $1 - \frac{1}{N^2 T^2}$ by setting $\delta = \sqrt{\frac{6 \log(NT)}{\mathbb{E}[d_s]}}$ and the second inequality follows from 43.

Combining 42 and 47 gives the result.

□

*Proof of Theorem 6.2.*

We will proceed similarly to the proof of Theorem 5.1, but we provide a complete proof here for clarity and completeness.

We have

$$R(S_\tau; \bar{v}_\tau) \geq R(S^*; \bar{v}_\tau) \geq R(S^*; v), \tag{48}$$

where the first inequality holds thanks to the definition of $S_\tau$ and the second inequality follows from Lemma B.1 with probability at least $1 - O(N^{-1}T^{-1})$.

Similar to the proof of Theorem 5.1, we can write

$$\mathrm{Reg}(T, \pi) = \mathbb{E}\left[ \sum_{\tau=1}^{\bar{\tau}} \left( 1 + v_{0d} + \sum_{j \in S_\tau} v_j \right) (R(S^*; v) - R(S_\tau; v)) \right].$$

And defining the epoch based regret be $\Delta R_\tau$, we can write it as

$$\Delta R_\tau = \left( 1 + v_{0d} + \sum_{j \in S_\tau} v_j \right) (R(S^*; v) - R(S_\tau; v)),$$

which will lead to

$$\text{Reg}(T, \pi) = \mathbb{E}\left[\sum_{\tau=1}^{\bar{\tau}} \Delta R_\tau\right].$$

We define event $\mathcal{E}$, for all $\tau$ as

$$\mathcal{E}_\tau := \bigcup_{i=1}^{N} \{|\bar{v}_{i,\tau} - v_i| > \Delta_\tau\} \cup \{|v_{0d} - \underline{v}_{0d,\tau}| > \Delta_\tau\}.$$

Conditioning on the event and by (48), we can write

$$\mathbb{E}[\Delta R_\tau] \leq (N+1)\mathbb{P}(\mathcal{E}_{\tau-1}) + \mathbb{E}\left[\left(1 + v_{0d} + \sum_{j \in S_\tau} v_j\right)(R_\tau(S_\tau; \bar{v}_\tau) - R(S_\tau; v))\,\mathbb{I}(\mathcal{E}_{\tau-1}^C)\right].$$

Then, by Lemma B.1 we have

$$\left(1 + v_{0d} + \sum_{j \in S_\tau} v_j\right)\left(R(S_\tau; \bar{v}_\tau) - R(S_\tau; v)\right)\mathbb{I}(\mathcal{E}_{\tau-1}^C)$$

$$\leq \left(1 + v_{0d} + \sum_{j \in S_\tau} v_j\right)\left(R(S_\tau; \bar{v}_\tau) - R(S_\tau; v)\right)$$

$$= \left(1 + v_{0d} + \sum_{j \in S_\tau} v_j\right)\left(\frac{\sum_{i \in S_\tau} r_i \bar{v}_{i,\tau}}{1 + \underline{v}_{0d} + \sum_{j \in S_\tau} \bar{v}_j} - \frac{\sum_{i \in S_\tau} r_i \bar{v}_{i,\tau}}{1 + v_{0d} + \sum_{j \in S_\tau} \bar{v}_j}\right.$$

$$\left. + \frac{\sum_{i \in S_\tau} r_i \bar{v}_{i,\tau}}{1 + v_{0d} + \sum_{j \in S_\tau} \bar{v}_j} - \frac{\sum_{i \in S_\tau} r_i v_i}{1 + v_{0d} + \sum_{j \in S_\tau} v_j}\right)$$

$$\leq \left(1 + v_{0d} + \sum_{j \in S_\tau} v_j\right)\left(\frac{\sum_{i \in S_\tau} r_i \bar{v}_{i,\tau}(v_{0d} - \underline{v}_{0d})}{\left(1 + \underline{v}_{0d} + \sum_{j \in S_\tau} \bar{v}_j\right)\left(1 + v_{0d} + \sum_{j \in S_\tau} \bar{v}_j\right)} + \frac{\sum_{i \in S_\tau} r_i(\bar{v}_{i,\tau} - v_i)}{1 + v_{0d} + \sum_{j \in S_\tau} \bar{v}_j}\right)$$

$$\leq \left(1 + v_{0d} + \sum_{j \in S_\tau} v_j\right)\left(\frac{K(v_{0d} - \underline{v}_{0d})}{\left(1 + \underline{v}_{0d} + \sum_{j \in S_\tau} \bar{v}_j\right)\left(1 + v_{0d} + \sum_{j \in S_\tau} \bar{v}_j\right)} + \frac{\sum_{i \in S_\tau}(\bar{v}_{i,\tau} - v_i)}{1 + v_{0d} + \sum_{j \in S_\tau} \bar{v}_j}\right)$$

$$\leq \left(1 + v_{0d} + \sum_{j \in S_\tau} v_j\right)\left(\frac{K(v_{0d} - \underline{v}_{0d})}{1 + v_{0d} + \sum_{j \in S_\tau} v_j} + \frac{\sum_{i \in S_\tau}(\bar{v}_{i,\tau} - v_i)}{1 + v_{0d} + \sum_{j \in S_\tau} v_j}\right)$$

$$= K(v_{0d} - \underline{v}_{0d}) + \sum_{i \in S_\tau}(\bar{v}_{i,\tau} - v_{i,\tau}).$$

Let $E_i$ be the number of periods that the product $i$ is offered. Then, the regret can be written as

$$\text{Reg}(T,\pi) \le \mathbb{E}\left[\sum_{\tau=1}^{\bar{\tau}}\left((N+1)\mathbb{P}(\mathcal{E}_{\tau-1}) + K\tilde{\Delta}_{0d,\tau} + \sum_{i\in S_\tau}\tilde{\Delta}_{i,\tau}\right)\right]$$

$$\le \mathbb{E}\left[\sum_{\tau=1}^{\bar{\tau}}\left(\frac{N+1}{NT^2} + K\sqrt{\frac{48\hat{v}_{i,\tau}\log(NT)}{|E_\tau(i)|}} + \frac{48K\log(NT)}{|E_\tau(i)|} + \frac{K\mathbb{E}[d_s]}{|E_i(\tau)|} + \frac{\sqrt{6\mathbb{E}[d_s]\log(NT)}}{|E_i(\tau)|}\right.\right.$$

$$\left.\left. + \sum_{i\in S_\tau}\left(\sqrt{\frac{48\hat{v}_{i,\tau}\log(NT)}{|E_\tau(i)|}} + \frac{48\log(NT)}{|E_\tau(i)|} + \frac{\mathbb{E}[d_s]}{|E_i(\tau)|} + \frac{\sqrt{6\mathbb{E}[d_s]\log(NT)}}{|E_i(\tau)|}\right)\right)\right]$$

$$\le \log(T) + K\sqrt{48T\log(NT)} + K(48 + \mathbb{E}[d_s] + \sqrt{6\mathbb{E}[d_s]})\log^2(NT)$$

$$+ 72\mathbb{E}\left[\sum_{i=1}^{N}\sqrt{v_i E_i\log(NT)}\right] + (48 + \mathbb{E}[d_s] + \sqrt{6\mathbb{E}[d_s]})\log^2(NT)$$

$$\le \log(T) + K\sqrt{48T\log(NT)} + (K+1)(48 + \mathbb{E}[d_s] + \sqrt{6\mathbb{E}[d_s]})\log^2(NT)$$

$$+ 72\sum_{i=1}^{N}\sqrt{v_i\mathbb{E}[E_i]\log(NT)},$$

where the third inequality holds due to $\bar{\tau} \le T$, $E_i \le T$ and the fourth inequality holds due to Jensen's inequality.

Since, $\sum_{i=1}^{N} v_i\mathbb{E}[E_i] \le T$, we conclude

$$\text{Reg}(T,\pi) \le \log(T) + K\sqrt{48T\log(NT)} + (K+1)(48 + \mathbb{E}[d_s] + \sqrt{6\mathbb{E}[d_s]})\log^2(NT)$$

$$+ 72\sqrt{NT\log(NT)}.$$

$\square$

# E   Experimental Details

We numerically evaluate our algorithms over 100 independently generated problem instances, with error bars representing standard errors in both Figure 1 and Figure 2.

The simulations were conducted on a server equipped with 4 Intel Xeon 6248 2.5GHz CPUs and 377 GB of RAM, running CentOS 7. The simulation code was developed in Python version 3.9.6.

