# OpenReview forum: "Online Learning of Delayed Choices"
_NeurIPS.cc/2024/Conference — NeurIPS 2024 poster_

### Official Review · Reviewer_3Qos · 2024-06-18

**Soundness:** 3
**Presentation:** 3
**Contribution:** 2
**Rating:** 6
**Confidence:** 3

**Summary:**

The paper studies a task of learning MNL parameters with delayed feedback. The authors study both the cases where feedback with extremely high delay is ignored or taken into account as well. They prove that for both settings the optimal regret is $\tilde{\Theta}(\sqrt{NT})$ where $T$ is the horizon and $N$ is the number of products. They also conducted experiments that support their theoretical findings.

**Strengths:**

1) Assuming that results are correct, the paper gives a first solution to a question that seems natural in the domain of online advertising.
2) The bounds are fairly tight.
3) The paper includes an extensive literature review.
4) The authors conducted experiments to support their theoretical findings.

**Weaknesses:**

1) I think that the main results could have been better presented. In the current formulation, it seems like there is no difference between the guarantees of DEMBA and PA-DEMBA, which makes the reader wonder why do we need both of them. It also makes it hard to understand what improvements can we hope to achieve in future work.
2) The setup seems kind of specific to this problem. It could have been interesting to know if one can define it more generally, and still use the same or similar solutions. This could have help the reader to conjecture if the same techniques may be used to solve other variations of the problem.

**Questions:**

Questions:
1) What is $\mathcal{S}$ in algorithm 1? The set of all assortments of size at most K?
2) $K$ is not mentioned in the upper bounds. They hold for all K?
3) What is the tradeoff between DEMBA and PA-DEMBA? The guarantees mentioned in Theorems 5.1 and 6.2 are identical.
4) In line 288, I am not sure how you reached the conclusion that "Our lower bound suggest an improvement on regret by considering the delay distribution via $\psi_\mu$." (there is also a typo here: suggest --> suggests). I don't understand how can a lower bound suggest an improvement to the upper bound. For example, $\Omega(1)$ is also a valid lower bound.

Suggestions:
1) The guarantees written in Theorems 5.1, 6.2 are identical. Consider showing the factors who make a difference between the results.
2) I think that the introduction is a bit lack of justifications or examples. For example, in line 23, you write "fall short in scenarios....". Why? Is there a line of work showing this?
3) A minor suggestion: In line 237, it is not clear why to use $\tilde{O}(\cdot)$ in this context. First, the lower bound has no logarithm factors in it so there's no point in concealing them. Second, $O(\sqrt{NT})$ can also be O(1), which is not the point of the lower bound. Therefore, I suggest writing "...better regret than $\Omega(\sqrt{NT})$".

Typos:
1) line 181: "at least at least".
2) Theorem 5.1: Add "Then" before "$\pi^{DEMBA}$".

**Limitations:**

Yes.

---

> ### Author Rebuttal · Authors · 2024-08-06
>
> First of all, we would like to extend our sincere thanks for your detailed review and constructive feedback. We have carefully considered your comments and have addressed them as follows:
>
> > W1) I think that the main results could have been better presented. In the current formulation, it seems like there is no difference between the guarantees of DEMBA and PA-DEMBA, which makes the reader wonder why do we need both of them. It also makes it hard to understand what improvements can we hope to achieve in future work.
>
> We appreciate your feedback regarding the presentation of the results. You are correct that the current use of $\tilde{O}$ notation may obscure the differences between the guarantees of DEMBA and PA-DEMBA. To address this, we will revise the manuscript to include all relevant factors in our regret bounds. This change will highlight the differences more clearly and provide a better understanding of the improvements that can be achieved with each algorithm.
>
> > W2) The setup seems kind of specific to this problem. It could have been interesting to know if one can define it more generally and still use the same or similar solutions. This could have helped the reader to conjecture if the same techniques may be used to solve other variations of the problem.
>
> Regarding the generalizability of our setup, we acknowledge that our work is indeed focused on the specific problem of delayed learning in the multinomial logit model. This specificity allows us to provide detailed analysis and guarantees for this particular setting, which is important in areas such as online advertising and recommendation systems. However, we believe that some of our techniques, particularly those dealing with delayed feedback, could potentially be adapted to other choice models or learning scenarios with delayed information. Future work could explore extending our approach to other discrete choice models or investigating how our methods for handling delay could be applied in different online learning contexts. While our current focus is on providing a thorough solution for the multinomial logit model with delayed feedback, we agree that exploring broader applications could be a valuable direction for future research.
>
> > Q1) What is $\mathcal{S}$ in algorithm 1? The set of all assortments of size at most \(K\)?
>
> $\mathcal{S}$ represents the set of all assortments of size at most K. We will make this clearer in the revised manuscript.
>
> > Q2) $K$ is not mentioned in the upper bounds. Do they hold for all \(K\)?
> > Q3) What is the tradeoff between DEMBA and PA-DEMBA? The guarantees mentioned in Theorems 5.1 and 6.2 are identical.
>
> $K$ and the difference between DEMBA and PA-DEMBA appear in logarithmic terms of our regret bounds, in the revised manuscript we will include logarithmic terms to clarify the distinctions between the algorithms.
>
> > Q4) In line 288, I am not sure how you reached the conclusion that "Our lower bound suggests an improvement on regret by considering the delay distribution via $\psi_\mu$." (there is also a typo here: suggest --> suggests). I don't understand how a lower bound can suggest an improvement to the upper bound. For example, $\Omega(1)$ is also a valid lower bound.
>
> The key point we were aiming to make is that our lower bound and upper bound do not match in terms of threshold terms $\psi_\mu$ and $\mu$ and this suggest a possible improvement on handling delay and/or threshold mechanism in our analysis. This distinction will also be clearer when we explicitly include the logarithmic terms in our regret guarantees in the revised manuscript.
>
> > S1) The guarantees written in Theorems 5.1, 6.2 are identical. Consider showing the factors who make a difference between the results.
>
> We agree with your suggestion. In the revised manuscript, we will explicitly detail the factors that differentiate the results, including constants and other relevant terms.
>
> > S2) I think that the introduction lacks justifications or examples. For example, in line 23, you write "fall short in scenarios....". Why? Is there a line of work showing this?
>
> We will modify this sentence with an example to illustrate that classical multi-armed bandit (MAB) models are not suitable for our setting. Specifically, MAB models require selecting one arm at each round, while in our case, we need to select multiple products (arms) to form an assortment. Moreover, adding or subtracting a product from the assortment affects the probabilities of selection for other products, which introduces a complexity not captured by traditional MAB models. This stochastic reward process and the impact of assortment changes on future decisions are better captured by our discrete choice model using the multinomial logit (MNL) framework. We will provide relevant justifications and examples to illustrate these points.
>
> > S3) A minor suggestion: In line 237, it is not clear why to use $\tilde{O}(\cdot)$ in this context. First, the lower bound has no logarithmic factors in it, so there's no point in concealing them. Second, $O(\sqrt{NT})$ can also be $O(1)$, which is not the point of the lower bound. Therefore, I suggest writing "...better regret than $\Omega(\sqrt{NT})$"
>
> Thank you for pointing out this typo. We will correct this and use $\Omega$ while calling our lower bound.
>
> > Typos
>
> We will correct the repetition of “at least” and we will add “Then” before “$\pi^{DEMBA}$" as suggested.
>
> We appreciate your feedback and believe these revisions will significantly improve the manuscript. Thank you for the opportunity to address these issues.

---

> > ### Comment · Reviewer_3Qos · 2024-08-08
> >
> > Thank you very much for addressing my comments and questions.

---

### Official Review · Reviewer_rHM8 · 2024-07-04

**Soundness:** 2
**Presentation:** 4
**Contribution:** 2
**Rating:** 4
**Confidence:** 4

**Summary:**

The authors consider the setting in which a business is required to select a set of options to a customer in order to maximize the generated revenue. This task is challenging as the options presented to a customer may interact with each other and alter the choice of the customer, and the feedback on the choice of the customer could be received by the business after a considerable delay. This class of tasks can be addressed in literature using Multinomial choice (MNL) models. The authors aim to address two challenges that arise in this scenario, namely the unknown MNL parameters and the delayed feedback. The authors consider two settings, *threshold* and *non-threshold*, and propose two algorithms to address them, DEMBA and PA-DEMBA, respectively.

**Strengths:**

The paper is clear and well-written. A deep analysis of related works is provided by the authors, clearly stating what gap in the literature the work aims to fill.
The work combines existing ideas to propose a solution to a problem that might have some significance to some members of the NeurIPS community.

**Weaknesses:**

W1) The definition of the "feedback observed by the seller", $o_{i,s,t}$ is inconsistent and seems also incorrect. At line 148, it is defined as:
$$
o_{i,s,t} = c_{i,s,t} a_{i,t},
$$
whereas in the proof sketch of Lemma 4.1 it is used as $o_{i,s,t} = c_{i,s,t} a_{i,s}$.
Both definitions seem to be incorrect, as in the first case (i.e., with $a_{i,t}$), $o_{i,s,t}$ evaluates to 1 considering the option chosen by the consumer at round $t$ instead of round $s$, in which the product was sold.
In the second case (i.e., with $a_{i,s}$), this issue is solve however the definition of $c_{i,s,t}$ makes it so that $o_{i,s,t}$ would evaluate to 1 at every round $t \ge d_s + s$ (under the condition that $d_s \le \mu$), potentially causing a choice of the consumer to count more than once towards the estimation of the preference.

W2) The proof sketch of Lemma 4.1 at line 188 (and in the appendix) seems to contain an error, as the conditions of $c_{i,s,t}$, which in the definition are in a logical AND, are split using a summation, which seems to be incorrect, and could invalidate all the subsequent steps of the proof.

W3) The observation that "each alternative can act as a substitute or competitors to others, impacting the customer's final decision" stated in Section 1 would have been better represented in the problem formulation with attraction parameters that depend on the options in the proposed set. Indeed, the provided formulation of the customer choice probabilities works fine, but cannot comprehensively capture product substitution dynamics.

**Questions:**

The reviewer would like the authors to address the reported weaknesses, and to respond to the following questions:

Q1) Can the authors provide a formal proof of the first passage of the proof of Lemma 4.1 at line 184:
$$\frac{ \sum_{\tau \in E_{\tau} (i) } \tilde{v}\_{i, \tau}}{| E_{\tau} (i) |} = \frac{\sum_{s=1}^{t_{\tau}^{end}} o_{i,s,t_{\tau}^{end}}}{| E_{\tau} (i) |},$$
as it is simply stated as a trivial definition however by applying the definition of $\tilde{v}_{i, \tau}$ it does not seem to be correct, considering also the remarks of W1).

Q2) Can the authors clarify how the epochs are defined?

Q3) Why have the authors not considered any of the algorithms reported in Section 2 in the experimental evaluation of the proposed algorithms?

**Limitations:**

No limitations

---

> ### Author Rebuttal · Authors · 2024-08-05
>
> First of all, we would like to extend our sincere thanks for your comments. We have taken due diligence to address the concerns raised and we believe your suggestions have greatly helped us in improving our manuscript.
>
> > W1) The definition of the "feedback observed by the seller", $o_{i,s,t}$ is inconsistent and seems also incorrect.
>
> Thank you for identifying this typo. We have updated line 148 as $o_{i,s,t} = c_{i,s,t} a_{i,s}$. We will explain the potential issue of multiple counting in the discussion of Q1.
>
> > W2) The proof sketch of Lemma 4.1 at line 188 (and in the appendix) seems to contain an error, as the conditions of \(c_{i,s,t}\), which in the definition are in a logical AND, are split using a summation, which seems to be incorrect and could invalidate all the subsequent steps of the proof.
>
> We can write the condition containing the logical AND as
> $$
> \mathbb{I}(d_s \le t-s \text{ and } d_s \le \mu) = \mathbb{I}(d_s \le \min(t-s, \mu)).
> $$
> In line 188, we split the expression based on the periods where the minimum attains a particular value. Therefore, our analysis is valid. We will add an explanation to make this step clear in our revised manuscript.
>
> > W3) The observation that "each alternative can act as a substitute or competitor to others, impacting the customer's final decision" stated in Section 1 would have been better represented in the problem formulation with attraction parameters that depend on the options in the proposed set. Indeed, the provided formulation of the customer choice probabilities works fine but cannot comprehensively capture product substitution dynamics.
>
> We agree that the multinomial logit (MNL) model has limitations. While the MNL model does not explicitly make attraction parameters dependent on the proposed set, the relative nature of the choice probabilities ensures that each item’s selection probability is influenced by the presence or absence of other items in the assortment. This captures a form of substitution effect. We acknowledge that more complex substitution dynamics could be modeled by making attraction parameters assortment-dependent. However, our current formulation offers a balance between capturing essential substitution behavior and maintaining model tractability. Exploring more nuanced substitution dynamics could indeed be an interesting direction for future research.
>
> > Q1) Can the authors provide a formal proof of the first passage of the proof of Lemma 4.1 at line 184:
> $$
> \frac{ \sum\_{\tau \in E\_{\tau} (i) } \tilde{v}\_{i, \tau}}{| E\_{\tau} (i) |} = \frac{\sum\_{s=1}^{t\_{\tau}^{end}} o\_{i,s,t\_{\tau}^{end}}}{| E\_{\tau} (i) |},
> $$
> as it is simply stated as a trivial definition; however, by applying the definition of $\tilde{v}\_{i, \tau}$, it does not seem to be correct, considering also the remarks of W1).
>
> Considering your comment in W1, we have refined our approach for representing the total observation count to avoid potential multiple counting. Hence, we modified the definition of $\tilde{v}\_{i, \tau}$ as
> $$
> \tilde{v}\_{i, \tau} = \sum\_{s=1}^{t^{end}\_\tau} o_{i,s, t^{end}\_\tau}
> $$
> and we can compute $\hat{v}\_{i, \tau}$ directly as
> $$
> \hat{v}\_{i, \tau} = \frac{\tilde{v}\_{i, \tau}}{| E\_{\tau} (i) |} = \frac{\sum\_{s=1}^{t^{end}\_\tau} o_{i,s, t^{end}\_\tau}}{| E\_{\tau} (i) |}.
> $$
>
> By this representation, we avoid any possible double counting and simplify the step asked in Q1.
>
> > Q2) Can the authors clarify how the epochs are defined?
>
> Epochs are defined by immediate no-purchase decisions. When an immediate no-purchase decision occurs, it closes an epoch and starts a new one. In line 164\&165, we mention that epochs are based on immediate no-purchase outcomes, however, we agree that this expression is not clear enough. We will make this definition explicit in our revised manuscript.
>
> > Q3) Why have the authors not considered any of the algorithms reported in Section 2 in the experimental evaluation of the proposed algorithms?
>
> Initially, we did not consider any algorithms from the literature in our experiments since our algorithm is the first one that considers delays and does not have a competitor in this regard. However, following your suggestion, we performed additional experiments comparing our algorithm (DEMBA) with MNL-Bandit (Agrawal et al.) and included them in the revised manuscript. We have also uploaded a file with these figures. We have three figures: starting with no delay and increasing the delay in the second and third figures. We observe that when there is no delay, the performance of MNL-Bandit and DEMBA are almost identical. However, as we increase the delay, the performance of MNL-Bandit deteriorates, clearly indicating that it fails to address delayed feedback effectively, whereas DEMBA continues to perform well.
>
> Once again, we would like to convey our sincere thanks for your thorough reading and your valuable feedback. We also hope that our revised version addresses all your comments.

---

### Official Review · Reviewer_dJHq · 2024-07-07

**Soundness:** 3
**Presentation:** 3
**Contribution:** 3
**Rating:** 7
**Confidence:** 4

**Summary:**

The authors studied the problem of learning with delated feedback under the Multinominal Logit model. Prior work in bandits with delayed feedback does not accommodate settings where multiple items can be offered simultaneously. The authors instead proposed two algorithms: DEMBA for thresholded setting where the seller discarded delay longer than a certain threshold, and PA-DEMBA when the threshold is infinity (when all delayed feedback are considered). Both algorithms achieve $O(\sqrt{NT})$ regret bound, and the authors provided matching lower bound up to a log term. Finally, the authors provided a set of numerical experiments to support their theoretical findings.

**Strengths:**

- The studied problem of delayed feedback in bandits, while not new, is interesting in the case where the seller can offer a slate of items to the customer at every round.

- The paper is well-written and easy to follow.

- The theoretical regret guarantee is provided with matching lower bound. These results and the numerical experiments provided a complete set of result for this setting.

- The proof-sketch provided offers good intuition to help understand the analysis and the result.

**Weaknesses:**

- The analysis did not attempt to learn the unknown delay distribution.

**Questions:**

- Does the problem setup change dramatically when the reward for each item in the assortment are not drawn i.i.d. For example, if the position of the item is correlated with the reward such that items put first in the list yields higher reward (or lower cost of browsing), does the current analysis still hold?

**Limitations:**

The authors have addressed the limitations of this work.

---

> ### Author Rebuttal · Authors · 2024-08-05
>
> Thank you for your detailed review and for your positive evaluation of our work. We appreciate your recognition of the strengths of our paper and your constructive feedback.
>
> > Does the problem setup change dramatically when the reward for each item in the assortment is not drawn i.i.d.? For example, if the position of the item is correlated with the reward such that items put first in the list yield higher reward (or lower cost of browsing), does the current analysis still hold?
>
> In our current setting, we use the multinomial logit model which assumes that the consumer choice behavior is solely affected by the subset of products being offered. We can extend this model by considering the choice probabilities, and therefore the expected reward of a product, is dependent to its position. We can define $\gamma_k$ as the visibility coefficient of the product in position $k$. We can also assume that $\gamma_k$ for each position is known to the learner. Indeed, $\gamma_k$ can be estimated from historical data. Then, the attraction parameter will be multiplied by the visibility parameter to calculate the expected reward, i.e., $r_i \frac{\gamma_{k(i)} v_i}{\sum_{j \in S} \gamma_{k(j)} v_j}$.
>
>
> Our analysis holds for this setting if we make an additional assumption that the customer viewed the whole assortment. Without this assumption, we cannot calculate choice probability by $\frac{\gamma_{k(i)} v_i}{\sum_{j \in S} \gamma_{k(j)} v_j}$ and we would need a different model for this case. In scenarios where customers may not view the entire assortment, our current model would require significant modifications to accommodate partial visibility. While other approaches such as cascading bandits (e.g., Combes et al., Craswell et al., Kveton et al.) explicitly model sequential item examination, integrating such models into our framework would be an interesting direction for future research rather than a direct solution within our current setup. With our whole-assortment viewing assumption and by using an appropriate argmax oracle (e.g., the work of Abeliuk et al. provides an efficient algorithm), we can cover the position-dependent scenario within our framework.
>
> We appreciate your insightful question and the opportunity to address it.
>
>
>
> A. Abeliuk, G. Berbeglia, M. Cebrian, and P. Van Hentenryck. Assortment optimization under a multinomial logit model with position bias and social influence. In 4OR, 14:57-75, 2016.
>
> R. Combes, S. Magureanu, A. Proutière, and C. Laroche. Learning to rank: Regret lower bounds and efficient algorithms. In Proc. of the 2015 ACM SIGMETRICS Int. Conf. on Measurement and Modeling of Computer Systems, 2015.
>
> N. Craswell, O. Zoeter, M. Taylor, and B. Ramsey. An experimental comparison of click position-bias models. In Proc. of the Int. Conf. on Web Search and Data Mining. ACM, 2008.
>
> B. Kveton, C. Szepesvári, Z. Wen, and A. Ashkan. Cascading bandits : Learning to rank in the cascade model. In Proc. of the 32nd Int. Conf. on Machine Learning, 2015.

---

### Official Review · Reviewer_KQSR · 2024-07-13

**Soundness:** 3
**Presentation:** 3
**Contribution:** 3
**Rating:** 5
**Confidence:** 2

**Summary:**

This paper works under the MNL bandit settings where the environment feedback is delayed, motivated by real-world application scenarios like e-commerce platforms, while balancing exploitation and exploration. For the two proposed algorithms, the authors provide corresponding theoretical analysis, resulting in regret upper and lower bounds. Experiments are also conducted to show the effectiveness.

Pros:

- The paper is generally well-organized with clear narratives and derivations. Indeed, delayed feedback is an important characteristic for modern recommender systems where users can remain neutral before disclosing their final preference towards recommendations.

- From my personal perspective, the theoretical analysis pipeline is novel, and the results look decent, with both the regret upper bound and lower bound presented.

Cons and questions:

- One question from my side is that for the current theoretical analysis, the regret bound mainly depends on the expectation of the delay, without modeling the skewness of the delay distribution $f_d$. In this case, I am wondering what the regret bound would look like if we take the skewness/variance of the distribution into account. For example, with $f_d$ being Gaussian, how will the variance interact with the final regret upper bound/lower bound? Is it possible to achieve a tighter regret bound when the distribution is decaying super fast compared to that of a long-tail distribution?

- Although the contribution of this paper mainly lies in the theoretical analysis perspective, it would be better if the authors could include more algorithms for comparison in their experiments. Some MNL bandit baselines from the related works section would be good.

**Strengths:**

Please see my comments above.

**Weaknesses:**

Please see my comments above.

**Questions:**

Please see my comments above.

**Limitations:**

Please see my comments above.

---

> ### Author Rebuttal · Authors · 2024-08-05
>
> First of all, we would like to extend our sincere thanks for your detailed review and for the insightful comments.
>
> > One question from my side is that for the current theoretical analysis, the regret bound mainly depends on the expectation of the delay, without modeling the skewness of the delay distribution . In this case, I am wondering what the regret bound would look like if we take the skewness/variance of the distribution into account. For example, with  being Gaussian, how will the variance interact with the final regret upper bound/lower bound? Is it possible to achieve a tighter regret bound when the distribution is decaying super fast compared to that of a long-tail distribution?
>
> Taking the skewness of the delay distribution into account can improve the regret upper bound, especially in the non-thresholded setting, although it will not change the asymptotic bound. However, this consideration is important in practice, as it can lead to tighter regret bounds under certain conditions. This would involve using Bernstein-type inequality, assumptions on tail characteristics(e.g. sub-exponential tails) or Gaussian assumption with known or unknown variance.
>
> In our revised manuscript, we will add a remark to explain that in practice, variations in the delay distribution with certain assumptions can be favorable. Specifically, for distributions with fast decay rates, such as Gaussian distributions, we can expect a better regret performance compared to long-tail distributions.
>
> Regarding the improvement mentioned in Remark 5.3 (reducing the $\mu$ factor in our upper bound), our current analysis does not admit improvement by incorporating the skewness of the delay distribution because currently we are using $\mu$ directly to bound one term in our analysis. We leave improving it as a future research direction.
>
> > Although the contribution of this paper mainly lies in the theoretical analysis perspective, it would be better if the authors could include more algorithms for comparison in their experiments. Some MNL bandit baselines from the related works section would be good.
>
> We agree and performed additional experiments comparing our algorithm (DEMBA) with MNL-Bandit (Agrawal et al.) and we will include them in our revised manuscript. We have also uploaded a file with these figures. We have three figures, we start with no delay and increase the delay in our second and third figure. We observe that when there is no delay, the performance of MNL-Bandit and DEMBA are almost identical. However, as we increase the amount of the delay the performance of MNL-Bandit deteriorates, clearly indicating that it fails to answer delayed feedback.
>
> Once again, we appreciate your insightful comments, which have significantly contributed to the enhancement of our manuscript. We hope these clarifications adequately address your concerns.

---

> ### Comment · Reviewer_KQSR · 2024-08-12
> **Thank you for the response**
>
> I would like to thank authors for your explanations. I will keep my current positive evaluation of your manuscript.

---

### Official Review · Reviewer_9pwP · 2024-07-13

**Soundness:** 3
**Presentation:** 3
**Contribution:** 3
**Rating:** 5
**Confidence:** 3

**Summary:**

The paper considers on online learning problem in the setting of discrete choice models with delayed feedback. The paper assumes a multinomial logit model where a decision maker has some (unknown) valuation v_i for item i. When presented with a menu S of choices, the agent chooses a single item from S such that the probability of selecting item i is proportional to v_i. In the online learning setup, at each time step t, the learner offers an assortment (menu) S_t and then agent chooses one item from the assortment. The paper considers a setup with delayed feedback, where the feedback regarding which item was chosen is delayed (according to an unknown distribution).

The paper considers two settings - (i) with censorship - where feedback delayed by more than some fixed deadline \mu is censored, and (ii) without censorship - where feedback can be delayed indefinitely (but expected delay is known to the learner). In both settings, the paper presents algorithms that obtain almost best possible regret \tilde O(NT) where N is the total number of items and T is the time horizon.
The algorithms are based on UCB and perform learning in epochs - i.e. offer the same assortment at all time steps throughout an epoch in order to reduce variance. An epoch is determined by times when the learner receives explicit negative feedback, i.e., the agent did not pick any item from the assortment.

**Strengths:**

- The paper considers online learning of discrete choice models in a general setting with delayed feedback. The setup is broadly applicable.

**Weaknesses:**

- I found the comparison with prior work a bit lacking. Since I am not directly familiar with works in this area, I would have appreciated more details about how the paper differs from prior work. In particular, Online learning with MNL choice models (but no delayed feedback) admits UCB based algorithms [Agrawal et al]. How much does the current work differ from that work? What additional technical complications are introduced by delayed feedback? Are they different from the challenges introduced by delayed feedback in other online learning settings (say classic MAB)?

**Questions:**

See above.

---

> ### Author Rebuttal · Authors · 2024-08-05
>
> Thank you for your thoughtful and constructive feedback on our paper. We appreciate your recognition of the strengths of our work and the detailed suggestions for improvement.
>
> Following your suggestion, we will expand the related work section in our revised manuscript to include a more comprehensive discussion of how our contributions differ from other available online learning methods with MNL choice models and other delayed learning settings. We also performed additional experiments to compare our algorithm with the literature.
>
> In particular, Agrawal et al. rely on unbiased estimations of attraction value i.e. $v_i$). However, in our setting with delayed feedback, we only observe biased samples which requires a different approach for estimation. We develop a concentration inequality in Lemma 1 that accounts for not-yet-observed choices. Without considering not-yet-observed choices, the high probability bounds would be invalid. Moreover, we performed additional experiments comparing our algorithm (DEMBA) with MNL-Bandit (Agrawal et al.) and we will include them in our revised manuscript. We have also uploaded a file with these figures.  We have three figures, we start with no delay and increase the delay in our second and third figure. We observe that when there is no delay the performance of MNL-Bandit and DEMBA are almost identical. However, as we increase the amount of the delay the performance of MNL-Bandit deteriorates, clearly indicating that it fails to answer delayed feedback.
>
> The challenges in our problem are similar to those in other online learning settings with delays in the higher level: dealing with the uncertainty due to not-yet-observed rewards. For UCB-based solutions, correcting bias due to delays is a common approach in the delayed bandit literature with stochastic delays. However, existing solutions for other bandit algorithms are not applicable to our problem due to the unique nature of assortment feedback in discrete choice models. Particularly, in MAB settings with delayed feedback, the primary concern is updating estimates of arm rewards. In our discrete choice model, we must maintain and update an assortment of items, creating a more complex interaction between choices and feedback. The expected reward structure is determined by an assortment of items, and we consider the bias at the item level. Delayed feedback affects both item value estimation and the assortment composition offered at each time step. Therefore, we developed a novel concentration inequality and used it to construct upper and lower confidence bounds of item attraction values that results in optimistic assortments.
>
> Once again, we thank you for your valuable feedback and for helping us improve our paper. We hope these clarifications adequately address your concerns.

---

### Author Rebuttal · Authors · 2024-08-07

We deeply appreciate the reviewers’ thoughtful and comprehensive comments and feedback. As per the suggestions of the review team, we’ve performed an additional experiment and are sharing the results in this file.

---

### Decision · Program_Chairs · 2024-09-25

**Decision:**

Accept (poster)

**Comment:**

There was unanimity in the fact that the problem studied by the authors is of broad interest to the NeurIPS community. Some reservations were expressed during the reviewing period, but after the discussions, no reviewers lamented that the authors were not able to address their major criticisms convincingly.
	Given the broad appeal of the topic and the overall positive opinions of the reviewers (which I also share), I recommend acceptance (poster) under the condition that the authors commit to incorporating all requested clarifications in the revised version.